# An Empirical MLR for Estimating Surface Layer DIC and a Comparative Assessment to Other Gap-filling Techniques for Ocean Carbon Time Series

Jesse M. Vance[1], Kim Currie[2], John Zeldis[3], Peter W. Dillingham[4], Cliff S. Law[1,5]

[1]Department of Marine Science, University of Otago, Dunedin, 9016, New Zealand
[2]National Institute of Water and Atmospheric Research – University of Otago Centre for Oceanography, Dunedin, 9016, New Zealand
[3]National Institute of Water and Atmospheric Research, Christchurch, 8011, New Zealand
[4] Coastal People Southern Skies Centre of Research Excellence, Department of Mathematics and Statistics, University of Otago, Dunedin, 9016, New Zealand
[5]National Institute of Water and Atmospheric Research, Wellington, 6021, New Zealand

*Correspondence to*: Jesse M. Vance (jesse.vance@icloud.com)

**Abstract.** Regularized time series of ocean carbon data are necessary for assessing seasonal dynamics, annual budgets, and interannual and climatic variability. There are, however, no standardized methods for filling data gaps, and limited evaluation of the impacts on uncertainty in the reconstructed time series when using various imputation methods. Here we present an empirical multivariate linear regression (MLR) model to estimate the concentration of dissolved inorganic carbon (DIC) in the surface ocean, that can utilize remotely sensed and modelled data to fill data gaps. This MLR was evaluated against seven other imputation models using data from seven long-term monitoring sites in a comparative assessment of gap-filling performance and resulting impacts on variability in the reconstructed time series. Methods evaluated included three empirical models: MLR, mean imputation, and multiple imputation by chained equation (MICE); and five statistical models: linear, spline, and Stineman interpolation, exponential weighted moving average and Kalman filtering with a state space model. Cross validation was used to determine model error and bias, while a bootstrapping approach was employed to determine sensitivity to varying data gap lengths. A series of synthetic gap filters, including 3-month seasonal gaps (spring, summer, autumn winter), 6-month gaps (centered on summer and winter) as well as bimonthly and seasonal (4 samples per year) sampling regimes were applied to each time series to evaluate the impacts of timing and duration of data gaps on seasonal structure, annual means, interannual variability and long-term trends. All models were fit to time series of monthly mean DIC, with MLR and MICE models also applied to both measured and modelled temperature and salinity with remotely sensed chlorophyll. Our MLR estimated DIC with a mean error of 8.8 µmol kg$^{-1}$ among 5 oceanic sites and 20.0 µmol kg$^{-1}$ for 2 coastal sites. The MLR performance indicated reanalysis data, such as GLORYS, can be utilized in the absence of field measurements without increasing error in DIC estimates. Of the methods evaluated in this study, empirical models did better than statistical models in retaining observed seasonal structure, but led to greater bias in annual means, interannual variability and trends compared to statistical models. Our MLR proved to be a robust option for imputing data gaps over varied durations and may be trained with either in-situ or modelled data depending on application. This study indicates that the number and distribution of data

gaps are important factors in selecting a model that optimizes uncertainty while minimizing bias and subsequently enables robust strategies for observational sampling.

## 1 Introduction

Despite continued policy development aimed at combating climate change and declines in carbon dioxide ($CO_2$) emissions by many countries over the last 10-15 years, global fossil fuel consumption continues to rise (Friedlingstein et al., 2020). We are now in unchartered territory, with anthropogenic carbon emissions over the last two and half centuries eclipsing that in the geological record of the past 66 million years, leaving the future of our marine and terrestrial ecosystems uncertain (Zeebe et al., 2016). Our ability to predict future conditions, affect policy and effectively manage climate change relies on understanding the feedbacks between climate, ecosystems, and biogeochemical cycles. To that end, the value of sustained time series observations has been well recognized for decades, as they are essential to characterizing processes, quantifying natural variability, identifying regime shifts and detecting long-term changes in our environment (Ducklow et al., 2009). Monitoring ocean carbon over the last three decades has revealed the decline in ocean pH concurrent with the uptake of 25% of anthropogenic $CO_2$ by the global ocean (Friedlingstein et al., 2020). Quantification of the ocean carbon sink and the impacts of ocean acidification remain actively researched given the significance of the ocean's role in controlling climate feedbacks as well as the ecological and economical importance of our marine systems (Kroeker et al., 2013; Devries et al., 2019; Krissansen-Totton et al., 2018; Bernardello et al., 2014). Ocean carbon programs have led to a growth in surface $pCO_2$ data from 250,000 global measurements in 1997 to 13.5 million in 2019; however, continuity and coverage of this inorganic carbon data in space and time remains a challenge for understanding seasonal and interannual variability (Takahashi and Sutherland, 2019; Takahashi et al., 1997).

### 1.1 Filling the gaps

Consistent sampling intervals for physical and biogeochemical parameters over several decades are critical for understanding ocean processes, establishing variability and detecting long-term changes (Henson et al., 2016). In addition to constraints arising from limitations in technology, logistics and funding, ocean science takes place in a particularly harsh environment where data loss is a common occurrence. Whether from equipment failure, cancelled field campaigns, budget cuts, or a global pandemic, gaps in time series are ubiquitous and must be appropriately filled in order to carry out various statistical analyses and modelling applications which require serially complete data sets.

Machine learning techniques such as neural network methods, regression trees, and random forests have been widely used to fill gaps in meteorological and some oceanographic data, including surface ocean $pCO_2$ (Laruelle et al., 2017; Sasse et al., 2013; Coutinho et al., 2018). While these methods are successful in the context of geospatial data, there remains little

standardization in methods for imputing data gaps in oceanographic time series, particularly carbonate chemistry, at monitoring sites where there are not sufficiently close neighboring values (in time or space) that can be leveraged. Linear interpolation and mean imputation are among the most common methods for handling missing data over short to moderate time scales (Reimer et al., 2017; Kapsenberg and Hofmann, 2016; Currie et al., 2011), but comparative assessment and validation of approaches overall is lacking. Gap-filling studies and standardization have been pursued in other terrestrial and atmospheric disciplines, such as eddy covariance carbon flux, solar radiation, air temperature, surface hydrology, and soil respiration (Moffat et al., 2007; Demirhan and Renwick, 2018; Zhao et al., 2020; Henn et al., 2013; Pappas et al., 2014), many of which focused on high temporal resolution data and imputing missing values over time scales from seconds to days. However it is important that the imputation method not only focuses on minimizing error but also minimizing bias, as the preservation of variance and trends is imperative for accurate analyses and understanding of climate (Serrano-Notivoli et al., 2019).

Here we present an empirical multiple linear regression (MLR) model for estimating site-specific DIC concentration in the surface ocean using remotely sensed data products to fill gaps in field measurement records. We compare this MLR approach to other commonly used and computationally inexpensive methods, including two empirical and five statistical methods. Using established carbonate time series from varied ecosystem types, we evaluate the sensitivity, error, and bias of these select methods and investigate the impacts of gap-filling on seasonal and interannual variability and long-term trends. Although the focus here is on DIC time series, the principles of this study should extend to other carbonate parameters.

## 2 Materials and Methods

### 2.1 Field data

We used data from the Bermuda Atlantic Time-series [BATS] (adapted from Bates et al., 2012), Carbon Retention In A Colored Ocean [CARIACO] (Astor et al., 2005; Astor et al., 2013), Firth of Thames [FOT] (adapted from Law et al., 2020), Hawaiian Ocean Time-series [HOT] (adapted from Dore et al., 2009), Kuroshio Extension Observatory [KEO] (Sutton, 2012a; Fassbender et al., 2017), Munida Time-series [Munida] (adapted from Currie et al., 2011), and Ocean Site Papa [Papa] (Sutton, 2012b; Fassbender et al., 2016). These time series present data describing significant ecological and environmental variability from different ocean basins and coastal regions (Fig. 1), which have been characterized in other studies (Bates et al., 2014; Fassbender et al., 2016; Fassbender et al., 2017; Zeldis and Swaney, 2018). Additionally, these time series have sufficient sampling frequencies and length of record to assess the monthly mean climatological conditions and seasonal cycle, so to allow inclusion of empirical imputation methods in this comparative assessment. Table 1 lists the site details including the carbonate

parameters measured, the duration of the time series, and the gap rate based on the expected sampling frequency for each of the seven sites.

All mixed layer temperature, salinity and dissolved inorganic carbon (DIC) data were averaged to monthly means for each time series site. For non-moored sampling sites with bottle sampling (BATS, CARIACO, HOT, Munida), monthly values were treated as the monthly mean condition. For each site the mixed layer depth was determined according to the temperature profile and a threshold of $\Delta T > 0.2$ °C relative to 10 m depth (De Boyer Montégut, 2004). For sites that did not measure DIC directly (Papa, KEO, FOT), the measured carbonate parameters were used with *in situ* temperature and salinity to calculate the DIC concentration and the uncertainty of calculation using the functions *carb* and *errors*, respectively within the R package *seacarb* (Jean-Pierre Gattuso et al., 2012; Orr et al., 2018) with $K_1$, $K_2$ from Lueker (2000); $K_f$ from Dickson (1979); and $K_s$ from Dickson et al. (1990); on the appropriate pH scale, where used, in R version 3.5.2 (R Core Team, 2020). DIC at Papa and KEO was calculated from measured $pCO_2$ and estimated total alkalinity (TA) based on the salinity-alkalinity relationships determined by Fassbender et al. (2016) and (2017) respectively. DIC at FOT was calculated from measured pH (SeaFet) and estimated TA based on the salinity-alkalinity relationship at that site (see supplemental material for more detail).

## 2.2 Remotely sensed and modelled data products

Monthly composites of satellite-derived surface ocean chlorophyll (O'Reilly et al., 1998) from MODIS data (Simons, 2020a) were paired with field data from each site except FOT. The mean surface chlorophyll was taken from a ~20 km$^2$ cell surrounding each of these sampling locations. For FOT, surface chlorophyll was estimated from monthly composite of VIIRS data (Simons, 2020b), with the mean from a ~ 4 km$^2$ cell surrounding the mooring used in this case given the greater spatial heterogeneity in this semi-closed coastal system. VIIRS also showed greater daily coverage of the FOT mooring location compared to MODIS, indicating a better representation of the monthly mean condition (see Supplemental Material).

Modelled monthly mean temperature and salinity profiles for each site were extracted from the GLORYS12V1 Global Ocean Physical Reanalysis Product (Global Monitoring and Forecasting Center, 2018; Fernandez and Lellouche, 2021; M. Drévillon, 2021). Temperature and salinity were averaged for the mixed layer depth in a ~20 km$^2$ cell surrounding each sampling location. GLORYS temperature and salinity were used only with empirical models where observations were either not available or synthetically removed for testing purposes. GLORYS temperature and salinity values were regressed against synchronized observations to quantify errors for each site (see Supplemental Materials).

## 2.3 Estimation of DIC with MLR

DIC, $pCO_2$ and other carbonate parameters have been successfully estimated in a variety of marine systems using multiple linear regression (MLR) approaches (Bostock et al., 2013; Velo et al., 2013; Hales et al., 2012; Lohrenz et al., 2018). In

addition, empirical estimates of pCO₂ using remotely sensed chlorophyll and sea surface temperature (SST) have proven useful for investigating seasonal and interannual dynamics across spatial gradients, particularly in coastal systems where sustained observations may be limited (Hales et al., 2012; Lohrenz et al., 2018). We investigated using an MLR model to estimate DIC from remotely sensed chlorophyll, SST and salinity in order to fill gaps in the seven monthly time series data. Parametric correlation matrices of DIC with remote chlorophyll, in situ SST and salinity showed significant linear correlation (Table 2), across most sites, with temperature having the strongest and most consistent correlation with DIC.

DIC at time $t$ can be estimated using MLR relationships described in the form of Equation 1.

$$E(DIC_t) = \alpha + \beta_1 Chl_t + \beta_2 T_t + \beta_3 S_t, \tag{1}$$

where $DIC$ has units of μmuol kg⁻¹, $Chl$ has units of mg m⁻³, $T$ has units of °C, and $S$ has units of psu and the coefficients $\alpha$ and $\beta_1$ through $\beta_3$ are the regression coefficients fit using a generalized linear model with a Gaussian error distribution and link function. The sensitivity to each predictor variable was assessed by selectively omitting chlorophyll, temperature, and salinity from the model fit.

The MLR model was also fit using GLORYS temperature and salinity data for each site to investigate its use for imputing gaps in observations, assuming no in situ measurements are available.

**2.4 Imputation of DIC time series**

Six general methods were compared for imputing DIC time series: classical, interpolation, Kalman filtering, weighted moving average (WMA) and regression and multiple imputation by chained equations (MICE). To apply the six methods, it must be assumed that the gaps in the time series are data 'Missing at Random', i.e. not missing systematically (Little, 2002). Given this assumption, these methods can be used to handle data gaps with limited biasing. This is suitable in our study where synthetic gaps are created using random number generators. However, this may not always be appropriate such as when data gaps are the result of systematic field site issues such as seasonal sea ice cover, season-specific sampling regimes, or seasonal biofouling.

The primary goal was imputing timeseries at monthly resolution to investigate variability and trends over seasonal, interannual and decadal timescales. Therefore, random sampling and persistence methods were not considered as these methods can lead to distortion of seasonal structure in the time series. Within the 6 methods chosen, 8 models were evaluated. These imputation models vary in complexity and flexibility and represent a range that have been widely applied to time series data, with 6 of the 8 models utilizing formalized packages (Demirhan and Renwick, 2018; Moritz, 2017). These methods limit overfitting and can be implemented with relative ease and low computational cost. Artificial data gaps were created as described below (Section 2.5) for the time series from each site in order to assess the performance of each method. In addition to the MLR model described by Equation 1, alternate models are described next.

The classical (and simplest) method applied was mean imputation, where missing values were replaced by the monthly climatological average. The climatological mean was taken as the monthly averaged means across the duration of the time series, which was over 1-2 decades in most cases.   Linear interpolation was used to estimate missing values by drawing a straight line between existing values in the time series and using the slope of each of these segments to determine the value of DIC at a time point(s) between known values. Spline interpolation utilized piecewise cubic polynomials to fit a curve with knots at $\xi_K$, $K = 1,2…k$, to the data, providing more flexibility with the ability to interpolate between each point of the training data. Stineman interpolation was developed to provide the flexibility of polynomials while reducing unrealistic estimations during abrupt changes in slope within the time series (Stineman, 1980) (see Demirhan and Renwick (2018) for algorithm details). Kalman filtering was implemented using a structural model. In this case a linear Gaussian state-space model was fit to the univariate time series by maximum likelihood based on decomposition (Demirhan and Renwick, 2018). A single weighted moving average model was evaluated. Missing values were replaced by weighted average of observations in the averaging window with size $k = \pm2$ and weighting was exponential such that the exponent increases linearly to the ends of the window, here ¼, ½… ½, ¼.

Multiple Imputation by Chained Equations (MICE), also known as fully conditional specification (FCS) and sequential regression multivariate imputation, was applied to time series data with artificial gaps and fit using the mice library (Van Buuren, 2011) in R version 3.5.2 (R Core Team, 2020), with function call *mice(data = TimeSeries$data, m = 5, method = "pmm", maxit = 20)*, where m is the number of multiple imputations,  method is predictive mean matching and maxit is the maximum number of iterations. This method progresses through the following steps: 1) missing values are filled by random sampling from the observations for a given variable; 2) the first variable with missing values is regressed against all other variables, while using only those with observed values; 3) moving iteratively, the remaining variables are regressed against the others but now including imputed values fitted by the regression models (White et al., 2011). This process is repeated according to the set iterations, in this case 20, to allow stabilization and convergence of the results. Regression models used in MICE allow for both linear and nonlinear relationships across variables, making this method very flexible.

## 2.5 Model performance and comparison

Each imputation model was evaluated using two schemes that assessed model performance and sampling sensitivity.

### 2.5.1 Cross validation

Leave one out cross validation (LOOCV) was chosen to assess the predictive error of the MLR model as well as the standard error for each imputation method. In this approach a single observation ($DIC_{t=1}$) is held out for validation while the remaining observations ($DIC_{t=2} … DIC_{t=n}$) are used for training the model. This process is repeated *n*-1 times, allowing each data point in the time series to be treated as both training data and testing data, thus maximizing the efficiency when the data sets are of

modest sampling size. Predicted DIC values and model parameters determined in each iteration were collated for the time series and performance statistics were evaluated on the total output.

### 2.5.2 Bootstrap sampling sensitivity

A bootstrapping approach was used to evaluate the sensitivity of the imputation models to the amount of data gaps in each time series. For each year of input data in the time series, artificial gaps were created by random removal of 1:8 monthly samples resulting in data gaps of 8.33%, 16.67%, 25.00%, 33.33%, 41.67%, 50.00%, and 66.67%. Random sampling was replicated 1000 times for each gap amount to ensure that an even distribution of sampling combinations was evaluated to assess the impacts of degree of data gaps on imputation error. Only years with 12 monthly samples were used to evaluate the sampling sensitivity in order to ensure consistency. It should be noted that most data sets used in this study do not have monthly mean data available for all years. Table 3 shows which years of data were used from each site and the distribution of years across sites.

### 2.5.3 Statistical performance metrics

The performance of each model was evaluated by comparing the predicted DIC values to the observed DIC measurements. The performance metrics included the coefficient of (multiple) determination ($R^2$) for indicating correlation; the root mean square error (RMSE), the relative root mean square error (RRMSE), and the mean absolute error (MAE) for establishing the distribution of individual errors; and the bias error (BIAS) for indicating bias induced on annual sums. Percent error (PE) and mean absolute percent error (MAPE) were used to evaluate particular metrics for assessing impacts of imputation on seasonal structure and long-term trends. Performance metrics were calculated according to Equations 2-8, where $o_i$ and $p_i$ denote the individual observed and predicted values respectively.

$$R^2 = \frac{\{\sum(p_i - \bar{p})(o_i - \bar{o})\}^2}{\sum(p_i - \bar{p})^2 \sum(o_i - \bar{o})^2} \tag{2}$$

$$RMSE = \sqrt{\frac{1}{N}\sum(p_i - o_i)^2} \tag{3}$$

$$RRMSE = \sqrt{\frac{\sum(p_i - o_i)^2}{\sum(o_i)^2}} \tag{4}$$

$$MAE = \frac{1}{N}\sum|p_i - o_i| \tag{5}$$

$$BIAS = \frac{1}{N}\sum(p_i - o_i) \tag{6}$$

$$PE = \left|\frac{p_i - o_i}{o_i}\right| \cdot 100\% \tag{7}$$

$$MAPE = \frac{100}{N}\sum\left|\frac{p_i - o_i}{o_i}\right| \tag{8}$$

## 2.6 Imputation effects on seasonal structure, interannual variability and long-term trends

To evaluate the impacts of imputation errors on seasonal structure, interannual variability and long-term trends we compared the observed and imputed time series using 8 synthetic gap schemes. Firstly, spring, summer, autumn, and winter seasonal gaps were evaluated by selectively removing 3-month windows from the DIC time series. Two longer 6-month sequential gaps scenarios were also used, one centered on winter and the other on summer. Lastly, two economical sampling schemes were evaluated, bimonthly (odd months only) and seasonal, in which only January, April, July and October were retained.

To evaluate the impacts on seasonal cycles and long-term trends, DIC was first normalized to the mean salinity ($S_0$) at each site per Equation 9.

$$nDIC_t = \frac{S_0}{S_t} \cdot DIC_t \tag{9}$$

The 8 imputation methods were applied to each of these 8 synthetic gap schemes for the full time series of nDIC at BATS, CARIACO, HOT, KEO, Munida, and Papa. Trends in the observed and imputed data were determined by least squares linear regression of the seasonally detrended time series, where the seasonal signal in each time series was removed according to Equation 10, following the methods in (Takahashi et al., 2009).

$$nDIC_{t,deseasoned} = nDIC_t - \left\{\overline{nDIC_t} - \overline{\overline{nDIC}}\right\}, \tag{10}$$

where $\overline{nDIC_t}$ is the climatological monthly mean and $\overline{\overline{nDIC}}$ is the climatological mean. FOT was not included in the evaluation because the time series of measured pH at this site is limited to 2015. To test the realistic application of the MLR and MICE models, it was assumed that measurement gaps resulted in missing observations of temperature and salinity along with DIC. While this may not always be the case, this allowed us to test using these empirical models to estimate DIC using a combination of remotely sensed chlorophyll data and modelled temperature and salinity in cases where all measurements are unavailable due to operational or logistical issues.

The PE of the time-regressed trends in nDIC were evaluated for each imputed time series compared to the observed trend in the data sets from each site. The mean seasonal cycle was evaluated as the monthly averages of the observed and imputed time series. Seasonal maximum and minimum concentrations of nDIC and their associated timing (which month) were compared. The seasonal amplitude, which was taken as the difference between maxima and minima of the climatological monthly means, and the interannual variability, which was taken as the standard deviation of the monthly means were also compared. Seasonal errors were combined according to Equation 11 for the purpose of comparing the overall impacts of each imputation method on seasonal structure.

$$PE(seasonal) = \sqrt{PE^2_{amplitude} + PE^2_{max} + PE^2_{min} + PE^2_{max\ timing} + PE^2_{min\ timing}} \tag{11}$$

## 2.7 Uncertainty budget

The sources of uncertainty accounted for here include measurement uncertainty, natural variability and the effect of monthly averaging, the effect of salinity normalization and the uncertainty associated with gap-filling. While individual measurement uncertainties may vary, measurement uncertainties across all sites in this study were treated as the following: salinity: 0.005 psu, temperature: 0.002 °C; pH: 0.05 units; pCO$_2$: 3 µatm; TA: 3 µmol kg$^{-1}$; DIC: 3 µmol kg$^{-1}$. These values were based upon reported uncertainties for in-situ temperature, salinity and pH (Sea-Bird Electronics, 2020, 2021) and pCO$_2$ (Jiang et al., 2008; Willcox et al., 2009; Johengen et al., 2009), and lab-based measurements of DIC and TA (Riebesell, 2011). Additional sources of uncertainty include: (1) estimation of monthly means, (2) estimation of TA from salinity (sALK), (3) calculation of DIC from sALK/pCO$_2$, (4) calculation of DIC from sALK/pH, and (5) salinity normalization of DIC (nDIC). Uncertainty associated with the calculation of DIC from other carbonate measurements combinations (e.g. sALK/pCO$_2$) was determined using the R package *seacarb* as described above. Uncertainty in TA estimated by salinity was taken as the $1.96 \times$ RMSE of the S-TA regression and propagated into DIC where needed.

Since the moored data here is averaged to monthly means for comparison with other observational time series, the uncertainty associated with this averaging must be accounted for. Additionally, the observational time series used in this study were treated as monthly means and the uncertainty associated with the natural variability at these sites must be estimated. The uncertainty associated with the averaging of monthly means was calculated by Equation 12.

$$u_{\bar{x}} = \frac{\sigma_{\bar{x}} t_{\bar{x}}}{\sqrt{n_{\bar{x}}}} \tag{12}$$

where $\sigma_{\bar{x}}$ is the standard deviation of the measurements within a month, $t_{\bar{x}}$ is the t-statistic, the ratio of the difference between the estimated and hypothesized value to the standard error, and $n$ is the number of measurements within a month (James, 2013). Uncertainty associated with monthly averaging was assessed directly for moored sites KEO and Papa. Because HOT represents a monthly sampled site, moored sensors data from 2016-2017 at WHOT (Terlouw et al., 2019) were used to evaluate the daily variability at this site and estimate the uncertainty associated with treating HOT samples as monthly averages. Uncertainty associated with monthly averaging for KEO, Papa, and WHOT ranged between 3-4 µmol kg$^{-1}$ for DIC and 0.03 – 0.05 psu for salinity and the upper limits of 4.00 µmol kg$^{-1}$ and 0.05 psu were applied as $u_{\bar{x}}$ in the combined uncertainty for DIC to all sites.

The uncertainty imposed from salinity normalization of DIC is calculated by taking the partial derivative of DIC with respect to salinity in Equation 8 and accounting for the uncertainty in salinity measurements and monthly averaging as given in Equation 13.

$$u_{nDIC_i} = \sqrt{\left(\frac{-S_0}{S_i^2}\right)^2 + u_{S_0}{}^2 + u_{S_i}{}^2} \tag{13}$$

Uncertainty in long-term trends was evaluated on the slope of the linear regression of the time series data according to Equation 14.

$$u_m = m\sqrt{\frac{1/_{R^2}-1}{n-2}} \tag{14}$$

where $m$ is the slope and $R$ is the coefficient of correlation. Combined uncertainty for imputed DIC values was evaluated by adding the sources of uncertainty in quadrature shown in equation 15.

$$u_{c(DIC)} = \sqrt{u_{DIC_i}^2 + u_{\overline{DIC}}^2 + u_{nDIC_i}^2 + RMSE_{method}^2} \tag{15}$$

## 3 Results

### 3.1 Seasonal cycles, interannual variability and long-term trends across sites

Box and whisker plots (Fig. 2) show the seasonal climatology and interannual variability for DIC and nDIC across the sites tested. The bar plots in Fig. 2 show the seasonal amplitude, which was taken as the difference between maxima and minima of the climatological monthly means, and the interannual variability, which was taken as the standard deviation of the monthly means. The amplitude of the seasonal cycle of DIC spanned 11.5 – 90.1 µmol kg$^{-1}$ across sites, while interannual variability ranged from 8.3– 22.6 µmol kg$^{-1}$. When the DIC is normalized to salinity the ranges of the seasonal cycles and interannual variability for nDIC become 12.7– 65.8 µmol kg$^{-1}$ and 7.6– 20.9 µmol kg$^{-1}$ respectively. The seasonal cycles, including amplitude, timing and interannual variability illustrate diversity among the test sites so enabling robust assessment of the empirical MLR model for surface layer DIC and other imputation methods. Figure 3 shows the long-term trends in DIC and nDIC time series from each site except FOT. Interestingly, with seasonal detrending, Papa uniquely exhibits a decline in DIC over the 10-year record used herein. Note here that BATS, CARIACO, and HOT time series were truncated to start at Sep 1997 when remotely sensed chlorophyll can be utilized in the empirical models (MLR and MICE) and compared to the other statistical approaches.

### 3.2 DIC estimation by MLR

Fig. 4 shows the performance of the MLR model to estimate DIC using the available time series data from each site ($N = 897$). The cross validated MLR exhibited an $R^2$ of 0.93 with an RMSE of 11.75 µmol kg$^{-1}$, RRMSE of 0.57%, MAE of 8.57 µmol kg$^{-1}$ and bias of 0.030 µmol kg$^{-1}$. The high $R^2$ and low error and bias indicate that the MLR model worked well for prediction of DIC from remotely sensed chlorophyll, and in situ temperature, and salinity across different ecosystems. The predictions and errors for the data from each site are provided in Table 4, which includes the means of the model coefficients and their standard deviations for the $N$ iterations of LOOCV per site.

The MLR performed best at Papa with a RMSE of 4.85 μmol kg$^{-1}$. This appears to be driven in part by low interannual variably and seasonal thermal stratification as discussed for reasons discussed below. The greatest error was associated with the CARIACO and FOT coastal sites, however, most of the predicted values still fell within 1% of observed DIC. When the sites were separated into oceanic (BATS, HOT, KEO, Papa and Munida) and coastal (CARIACO, FOT) categories, the RMSE was 8.75 μmol kg$^{-1}$ and 19.97 μmol kg$^{-1}$ respectively. When comparing the predictive accuracy of the MLR to the DIC variability at each site (Fig. 5), the interannual variability is strongly correlated (($R$) = 0.8532, p < .02) to the RMSE while the seasonal amplitude has no apparent impact (($R$)= 0.0771, p > .8), meaning the error in the predictions is most strongly related to interannual variability at each site.

To assess the sensitivity of the MLR to the predictor variables, the model was adjusted by selectively removing predictor variables and refitting the model. The changes in RMSE per site due to the omission of a given variable are shown as an anomaly in the tile plot of Fig. 6. BATS exhibited the greatest sensitivity to chlorophyll relative to other sites; FOT, HOT and KEO were relatively more sensitive to the effect of salinity; and temperature omission had the greatest impact for CARIACO, KEO, Munida, and Papa. The mean effects of variable omissions are given in Table 5, which indicates that collectively temperature had the greatest impact among the predictor variables on the predictive error. This was consistent with the expectations resulting from the correlation matrix provided in Table 2. The selective omission of predictor variables indicates that salinity contributes the most to the bias error although the bias error was low (<0.1) across all sites.

Comparing the GLORYS physical reanalysis data to the observations, the pooled RMSE was 0.68 °C for temperature and 0.18 psu for salinity with $R^2$ values of 0.9899 and 0.9841 respectively. The MLR performed similarly when GLORYS temperature and salinity values were used ($R^2$ = 0.9453, RMSE = 11.24 μmol kg$^{-1}$, RRMSE = 0.55%, MAE = 8.18 μmol kg$^{-1}$, and bias of 0.00000 μmol kg$^{-1}$; see the Supplemental Materials for more details).

### 3.3 Performance of imputation methods

Table 6 shows the pooled performance metrics for each cross validated model. These pooled results of the LOOCV indicate that each of the imputation models performed reasonably well with only 11% of all residuals exceeding 1% error and only 1 of 7424 estimated DIC values exceeded 5% error.

Overall, the MICE and MLR models exhibited the highest $R^2$ and lowest error (MAE, RMSE and RRMSE), followed by Kalman Filtering, Linear Interpolation, Exponential Weighted Moving Average, Mean Imputation, Stineman Interpolation, and Spline Interpolation in order of increasing RMSE. Mean exhibited the least amount of bias, while Spline Imputation exhibited the greatest amount of bias. Fig. 7 shows the kernel density curves of the residuals from the LOOCV of each imputation model with individual results from each site. Kernel density plots provide the probability distribution of the

residuals, where skewness and modalities (peaks) away from zero indicate biasing. Fig. 7 illustrates the error distribution varied greatly across sites when applying a selected model.


This considerable variability among the performance of each method across sites is further evidenced in Fig. 8. The tile colors in Fig. 8 indicate the RMSE and $R^2$ normalized to their pooled mean values for comparing the relative error and correlation across sites and methods. The individual cross-validated errors and $R^2$ values for each imputation method per site are given as the numerical value in each tile of the figure. Generally, Fig. 8 provides further evidence that CARICO and FOT exhibit the

greatest error overall, while KEO and Papa exhibit the lowest error. The $R^2$ panel in Fig.8 indicates that while some imputation errors may be low (<1%), they may still show poor correlation with observations. This is the case for statistical models at MUNDIA as well as mean imputation and spline interpolation models at HOT. The error and correlation across sites are consistent with the interannual variability shown in Fig.2 and with the MLR behavior shown in Fig. 5.

**3.4 Sampling sensitivity**

Sampling sensitivity was assessed by the RMSE for randomized artificial gaps totaling 8.33%, 16.67%, 25.00%, 33.33%, 41.67%, 50.00%, and 66.67%. The randomized approach resulted in a mixture of sequential and non-sequential gaps, while bootstrapping achieved equivalent representation of all months for each assessment. Fig. 9a shows boxplots of the RMSE for each imputation method as a function of percent of data missing at each site. Spline interpolation resulted in much greater magnitude and frequency of outliers and necessitated separate scaling. There was significant variability in both the

performance of different imputation methods within sites and within imputation methods across different sites. In general, mean imputation and MLR converge on a maximum error once data gaps reached 20-40%, whereas the error for other imputation models is positively correlated with the percent of data missing. While the performance of the cross validated Kalman filtering model did not differ greatly from the other interpolation methods, Fig. 9A indicates it leads to a greater number of outliers overall, in particular at BATS, KEO and Papa. Spline interpolation also resulted in a high number of outliers,

with the most extreme error over other methods. Fig. 9B shows the median error as a function of the percent of data missing with a loess fit. The general lack of a strong correlation shown by Mean imputation and MLR exhibit the least amount of sensitivity to the number of data gaps in the time series. The MICE model shows the highest level of sensitivity to the percent of data missing despite performing very well under the LOOCV and low numbers of data gaps.

**3.5 Time series gaps and trend assessment**

The imputed secondary time series synthesized with the 8 artificial gap scenarios, including sequential 3-month seasonal durations, 6-month durations centered on summer and winter, and bimonthly and seasonal sampling simulations are shown in the Fig. 10. Note that time series from each of the sites tested contained data gaps in the observations and synthetic gap scenarios were applied to the observed time series as-is. Extended gaps were observed at CARIACO (Apr 2001 – Feb 2002), KEO (Jan 2011 – Oct 2011), and Papa (Aug 2008 – May 2009).  Thirteen 3-month, three 4-month and one 5-month data gaps

present in the Muninda time series. Table 7 shows the number of observations for the total number of months in the time series at each site and the percent of data missing for each gap scenario tested.

Fig. 10 indicates a significant variability in the performance of each imputation method for the tested gap durations and timing within the datasets from each site. Note some outliers produced by spline interpolation were cropped in order to maintain

appropriate scaling of the y-axes. Overall, spline interpolation shows the highest propensity for creating outliers, as was also seen the in the assessment of sampling sensitivity. WMA shows a tendency for exaggerating seasonal minima and maxima, except in the cases of extended gaps, such as those seen at KEO and Papa. However, WMA remained within the observed range of annual seasonal cycles at Munida. Kalman filtering performed similarly to WMA. The empirical models (Mean, MLR, and MICE) better represent consistent seasonal cycles compared to other methods, as expected. However, these do not

perform as well when data deviate significantly from mean seasonal cycle, such as at HOT and CARIACO where the ratio of interannual variability to seasonal amplitude are high (84% and 46% respectively for nDIC). This is most clear in the high DIC concentrations observed at HOT during 2012-2013 and low DIC concentrations observed at CARIACO in 2003. KEO and Papa have the lowest ratio of interannual variability to seasonal amplitude (13%, and 14% respectively) and empirical models perform well here. This was consistent with the correlation between error and interannual variability evidenced by the LOOCV.

Fig. 11 shows the kernel density curves of the residuals between the infilled and observed nDIC values. The pooled residuals shown on the right-hand side of Fig. 11 indicate the time and duration of gaps has a significant impact on the error distribution.

Fig. 12 shows the kernel density curves of the residuals between the observed and reconstructed trends in nDIC over time for each site, method, and gap scenario. Trends from imputed time series that were significantly different than the observed trend

(taken here as a difference in trend that is beyond the uncertainty in the slope) are identified with a black asterisk in Fig. 12. Synthetic gap filters were applied by prescribed months across all sites rather than site-specific seasonal cycles and thus the impacts from each filter vary across sites. Generally, the mean imputation and MLR models led to reduced apparent trends across all sites by pushing the imputed values toward the climatological means. The exception to this was at Papa, where the bias was positive, in contrast to the apparent trend in the observations at that site. While this is inherent in mean imputation, it

is implicit in this MLR because it utilizes climatological relationships between the predictor variables rather than year-to-year variations. Linear and Stineman interpolation had the least impact on time series trends because values produced by these models are constrained to the range of the observations bracketing the gap and they tend more to preserve the trend as the observed values change through time. Except for KEO and Munida, Kalman and WMA models generally resulted in a reduced trends but with less error than the empirical models. The state space approach in the Kalman model attempts to describe the

dynamics through decomposition of the time series resulting in imputation values that are determined from prior observations, generally resulting predictions that are within the observed seasonal range. The tendency of the exponential weighting in the WMA is to overestimate when predicting values near maxima and minima (see Supplemental Material for additional figures). This is less apparent at Munida where the lower frequency of observations leads to weighting toward the annual means. This

balance in the WMA behavior explains its tendency for lower impact on the apparent trend. KEO exhibits both the strongest

trend in nDIC and largest seasonal amplitude and the Kalman and WMA models exaggerated the apparent trend here in all
gap scenarios. Spline interpolated values of the extended gap at CARIACO were well below the seasonal minima from previous
years in the time series and were extreme enough to inflate the trend in most of the gap scenarios.

The impacts on trends were greater for the 6-month gaps, bimonthly and seasonal scenarios than for the seasonal filters across

all models (see Supplemental Material for additional figures). This result is consistent with greater error being associated with
higher percentages of missing data, however, there was no direct correlation between imputation errors and the magnitude and
direction of changes in trends. The greatest impacts were observed when using mean imputation and MLR with the seasonal
sampling regime. This appears to be driven by the high percentage of data being replaced with climatological values.
Interestingly, MICE did not result in the same level of discrepancies with observed trends as the other empirical models. This

is likely due to the increased flexibility in the MICE model due to the inclusion of time fields (e.g. month as a predictor
variable) and the fact that the chained equation approach will allow for refitting throughout the time series allowing for year-
to-year variability in the relationships between predictor variables.

### 3.6 Seasonal cycles, annual means and interannual variability

The monthly means of the imputed time series and their associated uncertainties are shown in Fig. 13. These monthly series

more clearly illustrate the typical behavior of each imputation model described for each time series above. While deviations
from climatological monthly means are apparent across all sites, few of these fell outside of the uncertainty associated with
the observed monthly means, which is represented here by the combined sources of uncertainty in measurements and
calculation of the monthly mean nDIC and does not include the interannual variability of the monthly means.

The effects of imputation on the seasonal maxima and minima, their respective timing and amplitude are shown in Fig. 14,

which also includes residuals for interannual variability, annual means and the combined seasonal error pooled across sites.
Two-way ANOVA of each of these seasonal residuals indicated that the distribution of errors among the different models was
significantly different for seasonal amplitude, maxima, minima, while the difference between gap scenarios was significant
for the timing of seasonal minima. The combined seasonal error was significantly different among both imputation models
and gap scenarios. The residuals of annual means were also significantly different among both imputation models and gap

scenarios, while only model selection resulted in significantly different interannual variability.

The weakening of seasonal amplitude from linear imputation methods is evident in the residuals for all gap scenarios, as is the
tendency for the Kalman and WMA models to increase seasonal amplitude. The autumn gap filter resulted in the greatest

amount error in seasonal amplitude. This was driven by the larger residuals in the seasonal minima since most of the test sites
experience seasonal minima during autumn months. This also affected the timing of seasonal minima with residuals of up to

3 months.  The distribution of the seasonal residuals among the imputation models for the 6-month winter gap were similar to those for the autumn gap, although the residuals for seasonal minima, maxima and amplitude were largest with the 6-month winter gap filter.


The combined seasonal errors indicate that next to mean imputation, MLR does the best out of the other methods tested to retain the climatological seasonal structure observed at each site. The combined seasonal MAPE was 7.2% MLR, followed by 14.2% for spline interpolation, 15.1% for MICE, 19.2% for Stineman, 19.8% for Kalman, 19.9% for linear interpolation, and 21.1% for WMA. The autumn gap filter resulted in a combined seasonal MAPE of 20.9%. This was just over double that of
all other seasonal gap filters which resulted in error that ranged 8.8 – 9.9%.  The seasonal error was largest for the 6-month winter gap with a median error of 26.4%. Interestingly, the bimonthly sampling regime resulted in a seasonal MAPE of 16.8%, which was greater than 6-month summer gap (15.1%) and the spring, summer, and winter seasonal gaps, despite greater dispersed data coverage across seasons compared to these other scenarios. The seasonal MAPE for the seasonal sampling regime was 12.7% and lower than that exhibited by the more frequently bimonthly sampling.


The pooled residuals for annual means were mostly normally distributed about a median of 0 $\mu$mol kg$^{-1}$ with some biasing. When looking at the MAPE the seasonal gap filters and bimonthly sampling regime led to small errors in annual means of 0.1% while the 6-month gaps and seasonal sampling regime were 0.15-0.16%. When the errors are broken down by model selection, the empirical models showed the greatest deviation from the annual means, with mean imputation having a median
error of 0.16%, MLR 0.16%, and MICE performing slightly better at 0.13%. These were followed by Kalman 0.12%, spline interpolation and WMA at 0.11%, Stineman and linear interpolation at 0.08% in decreasing order.

The pooled residuals for interannual variability exhibited significantly more biasing and errors. The MAPE of interannual variability for each gap scenario correlated with the percent of missing data for each gap filter. The seasonal filters had errors
of 7.9-9.3%, followed by bimonthly 12.9%, 16.3% for the 6-month winter and summer gaps, and the seasonal filter at 19.1%. The error in interannual variability imposed by the models were highest for mean imputation at 22.5%, followed by spline interpolation 19.3%, WMA 13.7%, Kalman 12.0%, Stineman 9.6%, linear interpolation 9.3%, MLR 10.7% and MICE at 7.9%.

## 4 Discussion

### 4.1 MLR estimation of DIC

The development of remote sensing and MLR-based approaches for carbonate chemistry have been used extensively for extrapolating over broad spatial and temporal scales to investigate regional to basin scale phenomena (Bostock et al., 2013; Hales et al., 2012; Evans et al., 2013; Lohrenz et al., 2018; Juranek et al., 2011; Alin et al., 2012). Remote sensing applications have focused primarily on predicting $pCO_2$ and estimating air-sea flux in coastal waters to better understand the seasonal and

spatial heterogeneity of carbon sources and sinks and their implications for regional and global carbon budgets (Hales et al., 2012; Lohrenz et al., 2018). Many MLR models that predict carbonate parameters have been developed using large observational data sets that include either dissolved oxygen ($O_2$) (Juranek et al., 2009; Kim et al., 2010; Alin et al., 2012; Bostock et al., 2013) or nitrate ($NO_3$) (Evans et al., 2013) as a predictor variable along with temperature and salinity. MLR models that incorporate $O_2$ and $NO_3$ can perform particularly well in coastal environments where ecosystem metabolism has a dominant effect carbonate chemistry (Alin et al., 2012, {Juranek, 2009 #1264)). However, there are currently no remotely sensed $O_2$ and $NO_3$ data products and the chances of glider or float data being available at a given time series site to coincide with a gap in carbonate measurements are limited. The MLR model presented herein serves as a method for imputing missing DIC values in time series. This MLR may be trained and implemented using remotely sensed chlorophyll with in-situ temperature and salinity. However, for cases when in-situ temperature and salinity are concurrently unavailable during gaps in DIC observations, model-based estimates of temperature and salinity may be used as we have shown here with the Mercator Ocean Global Reanalysis (GLORYS). Additional data product options could include the Hybrid Coordinate Ocean Model (HYCOM), the Climate Forecast System Reanalysis (CFSR), and the Bluelink Reanalysis (BRAN), with assessment for a given location and included in the uncertainty budget (De Souza et al., 2020). Satellite-based estimates of sea surface temperature and salinity may also be considered although remotely sensed salinity typically has a larger error than the GLORYS data presented here when compared to observations (Wang et al., 2019).


The variability in the MLR model coefficients indicated that the relationships between DIC, chlorophyll, temperature and salinity were location-specific and cannot be spatially extrapolated to different water masses and ecosystems. This was indicated by the variability seen among the correlations of predictor variables to DIC across sites and clearly evidenced by the differences in model performance between the coastal sites (FOT and CARIACO) and the oceanic sites. However, when the MLR was trained with sufficient observations to capture the seasonal cycle, it can predict DIC with error that was far less than the natural variability over seasonal and interannual time scales and was typically on the order of, or better than the variability on monthly time scales. The RMSE of 4.85 – 10.67 µmol kg$^{-1}$ at the oceanic sites is consistent with other MLR studies which have ranged from ~4-11 µmol kg$^{-1}$ (Evans et al., 2013; Juranek et al., 2011; Bostock et al., 2013), while the RMSE at coastal sites (FOT and CARIACO) of approximately 20 µmol kg$^{-1}$ is larger than exhibited in a California Current study (Alin et al., 2012). The Alin study, like others (Juranek et al., 2009; Juranek et al., 2011), estimated DIC based on $O_2$ and density, incorporating a multiplicative relationship. While $O_2$ may improve the performance of MLR approaches, particularly in biologically active coastal environments, the MLR model here only utilized remotely sensed chlorophyll and temperature and therefore only applied to the surface layer. $O_2$ and $CO_2$ may become decoupled in the surface layer due to varying time scales for air sea gas exchange, making $O_2$ a less reliable predictor variable for surface concentrations of DIC (Juranek et al., 2011). Despite somewhat higher RMSE in coastal environments relative to the results of Alin et al. (2012), the MLR model here exhibited predictive error that is still less than 1% at such sites. With the mean performance among oceanic sites being 8.75

μmol kg$^{-1}$ and within the "weather" requirements adopted by the Global Ocean Acidification Observing Network, we contend that this is an acceptable approach for temporal interpolation (Newton, 2015).

**4.2 DIC time series imputation**

Despite the pervasiveness of gaps in climatological data aimed at understanding the ocean carbon cycle, there is limited evaluation of errors and bias in reconstructed time series due to gap-filling methods outside of the spatiotemporal interpolation in surface ocean pCO$_2$ datasets (Gregor et al., 2019). The MLR presented herein was developed as an empirical method toward constructing gap-filled regularized DIC time series, specifically for investigating seasonal and interannual variability in the carbon cycle within the surface layer. A thorough characterization of implementing this model beckoned the comparison to

other commonly used techniques and provided the opportunity to investigate the temporal and seasonal impacts of gap-filling.

Cross validation of the imputation models evaluated in this study indicated that each of these models have reasonably low (typically <1%) error when imputing a single value at monthly timescales. This was similar to other comparative gap-filling studies in the fields of soil respiration, net ecosystem exchange, and solar radiation, which focused on higher temporal

resolution data and imputing missing values over time scales from seconds to days (Moffat et al., 2007; Zhao et al., 2020; Demirhan and Renwick, 2018). For the assessment of annual budgets in the studies of Zhao et al (2020) and Moffat et al (2007), the error associated with the imputation methods was similar to the uncertainty in the fluxes across sites (Lavoie et al., 2015). As a result, the choice of imputation model yielded limited improvement on the accuracy of budget estimates. Similarly we found that the MAPE was under 0.2% for the annual means calculated from imputed time series, which was less

than the relative uncertainty for annual mean concentrations in surface layer DIC were on the order of 0.5-1%. However, Fig. 14 shows this can be biased positively or negatively depending on imputation method. While imputation resulted in limited error in annual means, there were significant impacts on the interannual variability, which ranged from 8-19%. Such errors would have a direct impact on the time of emergence in detecting trends (Sutton et al., 2019; Turk et al., 2019). Furthermore, our evaluation of reconstructed DIC time series with synthetic gaps showed that selection of imputation method can have

significant effects on the calculated timing, magnitude and structure of seasonal variability as well as longer temporal trends. The timing and duration of data gaps are important considerations, as are the research objectives for a given study and whether seasonal or climatic variability are more heavily weighted.

The empirical models evaluated in this study performed better than others selected here to maintain all aspects of the seasonal

structure. Mean imputation, by definition, maintains the climatological seasonal structure perfectly. However, year-to year this may lead to bias in the seasonal amplitude up or down relative to the temporal position in the time series and any long-term trend. This is apparent in interannual variability of reconstructed timeseries showing a positive bimodal distribution of the residuals for mean imputation (see Fig. 14), indicating larger error associated with a higher percent of missing data.

When looking at the combined seasonal error of each model pooled for all gap scenarios, MLR performs better than twice as well as all remaining methods and was the only model (other than mean imputation) with a median error under 10%. Looking at the individual imputed time series, the MLR generally tracks closely with mean imputation but with added interannual variability. This leads to less error compared to mean imputation as also seen in the distribution of residuals (see Fig. 11). The MICE model showed considerably more variability in its prediction of DIC values, leading to higher error with a wider distribution. This was likely due to the MICE method refitting regression models along the time series, whereas the MLR, as presented here, is fit once using the entire time series.

While mean imputation and MLR provide the best options of the models evaluated here for maintaining the seasonal structure in the time series, it is at the sacrifice of maintaining the observed trend. These two models led to the greatest discrepancies between observed and reconstructed trends. Both models act to weaken the trend, pushing toward the climatological mean; and this becomes more apparent with increasing data loss. Linear and Stineman interpolation models generally do well to maintain the observed trend in the time series due to them constraining infilled values between existing observations along the trending time series. This is at the sacrifice of maintaining seasonal structure as is clearly evidenced in Figs. 13 & 14. Even under the bimonthly sampling regime, these interpolation methods lead to a lower seasonal amplitude and this impact is worsened by longer duration gaps. Spline interpolation, WMA, Kalman filter and MICE models exhibit inconsistent impacts on trends across sites and varied gaps. WMA and Kalman performed best at Munida with limited bias, while MICE performed well during some gap scenarios at BATS (spring, summer, and 6-month summer gap) and KEO (spring, winter, seasonal); likewise for spline interpolation at BATS (spring, seasonal) and HOT (spring, summer, autumn, 6-month summer gap, and seasonal).

The impact on trend assessment does not appear correlated with the mean imputation error, bias, or mean seasonal errors; rather, visual inspection of the imputed time series in Fig. 10 appears to indicate that the timing of data gaps relative to how a selected model typically responds to such a gap, dictates the bias error for that gap. This bias error may then be exaggerated for longer durations and accumulate in the reconstructed time series and ultimately impart bias on the trend, even if the mean errors remain small. While using static month-based gap filters in our assessment, it also appears that in some cases interannual variability in the seasonal cycle changed the gap filter window. For example, linear and Stineman interpolation applied to the 6-month winter gaps at KEO 2008-2009 and 2015-2016 lead to a higher mean DIC concentration over these windows, leading to lower trend in these reconstructed time series than was observed. Additionally, spline interpolation was biased at HOT using the winter gap filter due to the splines exaggerating some of the seasonal transitions 2004 – 2009. The seasonal cycles 2006 – 2009 were further exaggerated using the 6-month winter gap filter leading to bias in the other direction. The correlation between trend error and imputation performance presents an area for further investigation.

One-way ANOVA indicated that the distribution of RMSE resulting from each of the gap scenarios were significantly different for each of the imputation models tested, further indicating the importance of the timing and duration of data gaps. Of the four

seasonal filters, spring data gaps had the least impact (lowest error), while autumn data gaps had the most. Given that these correspond to the seasonal maxima and minima respectively, it is interesting that selected imputation models are generally better at predicting the seasonal highs rather than lows. Errors associated with seasonal minima were further exacerbated by the long 6-month winter gap tested, whereas the 6-month gap centered in summer had errors that were on the order of other seasonal 3-month gaps. Collectively these results can help guide strategy for both sampling and the handling data gaps.


Bimonthly and seasonal sampling regimes provide economical options for data collection. The median RMSE associated with the bimonthly and seasonal sampling regimes were 10.4 µmol kg$^{-1}$ and 10.7 µmol kg$^{-1}$ respectively. These were less than the errors associated with summer (11.3 µmol kg$^{-1}$) and autumn (12.1 µmol kg$^{-1}$) gap filters and similar to the spring (10.7 µmol kg$^{-1}$) and winter RMSE (10.4 µmol kg$^{-1}$). This result is encouraging despite the bimonthly and seasonal sampling regimes

equating to twice as much data loss compared to the seasonal filters. These sampling regimes also impart similar results with respect to maintaining seasonal structure; although, bimonthly sampling leads to greater variance. Bimonthly sampling resulted in a median RMSE for annual means of 4.0 µmol kg$^{-1}$, equal to a typical measurement uncertainty. This was only slightly higher for seasonal sampling at 5. µmol kg$^{-1}$. The RMSE for interannual variability for these sampling regimes are less than 3 µmol kg$^{-1}$. These results are promising to indicate that these economic sampling regimes can capture the seasonal cycle with

reasonable uncertainty. However, it must be noted that these pooled errors include the performance and low errors of mean imputation and MLR and these empirical models require multiple years of data to adequately train. Uncertainty of annual and seasonal data based on these regimes would be higher.

The results presented here indicate that care should be taken when considering what method to use to fill data gaps in ocean

carbon time series, with criteria for selection including the percent of missing data, gap lengths and site characteristics. Of the methods we tested, the empirical models performed better than statistical models evaluated in this study with respect to imputation error and retaining seasonal structure. Mean imputation provides a stable and straightforward approach to filling longer gaps but leads to greater biases in annual budgets, interannual variability and long-term trends compared to the other methods evaluated in this study.


MICE appeared to be well suited to environmental time series data that have covariate parameters such as the correlation between DIC, chlorophyll, temperature and salinity. This could be extended to other nutrients such as phosphate and nitrate as well as dissolved oxygen in order to train the models used in MICE. MICE also offers the opportunity to impute data gaps over multiple variables in larger time series data sets. MICE does well to limit biases and did best to reproduce interannual

variability across the sties tested. MICE performed very well during cross validation but exhibited higher RMSE compared to MLR when reconstructing the time series, perhaps due to its greater sampling sensitivity shown in Fig. 9.

     Our MLR model provides a stable option that performs well over all rates of data missingness once it is sufficiently trained with field data. This MLR performed equally well using GLORYS reanalysis temperature and salinity data. This approach

provides the benefit of utilizing remotely sensed and modelled data products in the absence of covariate field data. The low error and uncertainty associated with this MLR approach show promise. Allowing the model to update the fit and coefficients for the predictor variables over the time series may help reduce biasing of temporal trends while maintaining the ability to retain seasonal structure. This MLR has potential to be trained with field data over broader spatial extents to assess regional carbon cycles.

**5 Conclusions**

     This study provides the first comparative assessment of several common gap-filling methods which are easy to implement and computationally inexpensive that may be applied to ocean carbon time series. Regularized carbonate time series data are necessary for understanding seasonal dynamics, annual budgets, interannual variability and long-term trends in the ocean carbon cycle and changes to the ocean carbon sink, which are of particular importance in the face of global climate change.

Our assessment indicates that the amount and distribution of gaps in the data should be a determining factor in choosing an imputation method that optimizes uncertainty while minimizing bias. Imputed values, however, cannot be treated as measurements and the uncertainty of imputation methods must be included in the overall uncertainty budget of broader ocean carbon analyses. The results presented above indicate the performance and behavior of select empirical and statistical approaches and the methods used provide a simple approach for estimating uncertainty of DIC predicted by a given imputation

method.

     This study provides evidence that DIC can be estimated with an empirical MLR approach that utilizes remotely sensed chlorophyll and may be trained with either in-situ or modelled temperature and salinity depending on the intended application. This method performs consistently well across 7 disparate ecosystems in oceanic and coastal environments, but the model

coefficients are unique to the water mass and ecosystem and further study is needed to assess the spatial extent over which regional extrapolation is still valid. However, when using this method to impute data gaps in carbonate time series, it performs better than several options, particularly for larger gaps. We conclude that when trained with sufficient field data (e.g., captures the seasonal cycle and some interannual variability), this empirical MLR method predicts DIC with acceptable accuracy from remotely sensed data and provides the most robust option from those we compared for imputing gaps over a variety of data

gap scenarios.

**Data and Code Availability**

The data sets and processing code used for the analyses in this study can be found under the figshare project: An Empirical MLR for Estimating Surface Layer DIC and a Comparative Assessment to Other Gap-filling Techniques for Ocean Carbon TimeSeries

(https://figshare.com/projects/An_Empirical_MLR_for_Estimating_Surface_Layer_DIC_and_a_Comparative_Assessment_to_Other_Gap-filling_Techniques_for_Ocean_Carbon_Time_Series/100349)

**Author Contributions**

JV performed conceptualization, data curation, software development, formal analyses, visualizations, and preparation of the manuscript. JV developed the MLR, with PD and KC providing support on methods used in the tests carried out in this study.

Supervision was performed by KC and CL. KC and JZ contributed to curation of select data. All authors contributed to review and revision of the manuscript.

**Competing Interests**

The authors declare they have no conflicts of interests.

**Acknowledgments, Samples, and Data**

The authors thank the following for long-term contributions to ocean carbon time series and access to high quality data: the Institute for Marine Remote Sensing team for making data from the Cariaco Basin publicly available; the members of the Bermuda Institute of Ocean Sciences for making data from the Bermuda Atlantic Time Series publicly available; the School of Ocean and Earth Science and Technology at the University of Hawaii Manoa for making data from the Hawaiian Ocean Time-series publicly available; NOAA's Pacific Marine Environmental Laboratory for making data from Ocean Station Papa

and the Kuroshio Extension Observatory publicly available; and to New Zealand's National Institute for Water and Atmospheric Research for providing data from the Munida Time Series and the Firth of Thames. The research presented herein was supported financially by NIWA Research Scholarship Grant C 17959. This publication is based in part upon Hawaii Ocean Time-series observations supported by the U.S. National Science Foundation (NSF) under Award #1756517. This publication is based in part upon the CARIACO Ocean Time Series program observations supported by the NSF, the U.S. National

Aeronautics and Space Administration (NASA), and Venezuela's Fondo Nacional de Ciencia, Tecnología e Innovación (FONACIT). This publication is based in part upon Bermuda Atlantic Time-series Study observations supported by the NSF under Award # 0326885. This publication is based in part upon the Kuroshio Extension Systems Study observations supported by the U.S. National Ocean and Atmospheric Administration (NOAA) and the Japan Agency for Marine-Earth Science and

Technology (JAMSTEC)'s Institute of Observational Research for Global Change (IORGC). This publication is based in part upon Ocean Station Papa observations supported by NOAA, the NSF and University of Washington. These data were provided by NOAA's Center for Satellite Applications & Research (STAR) and the CoastWatch program and distributed by NOAA/NMFS/SWFSC/ERD. This publication is also based in part upon observations from the WHOI-Hawaii Ocean Timeseries Site (WHOTS) mooring, which is supported by the National Oceanic and Atmospheric Administration (NOAA) through the Cooperative Institute for Climate and Ocean Research (CICOR) under Grant No. NA17RJ1223 and NA090AR4320129 to the Woods Hole Oceanographic Institution, and by National Science Foundation grants OCE-0327513, OCE-752606, and OCE-0926766 to the University of Hawaii for the Hawaii Ocean Time-series.

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

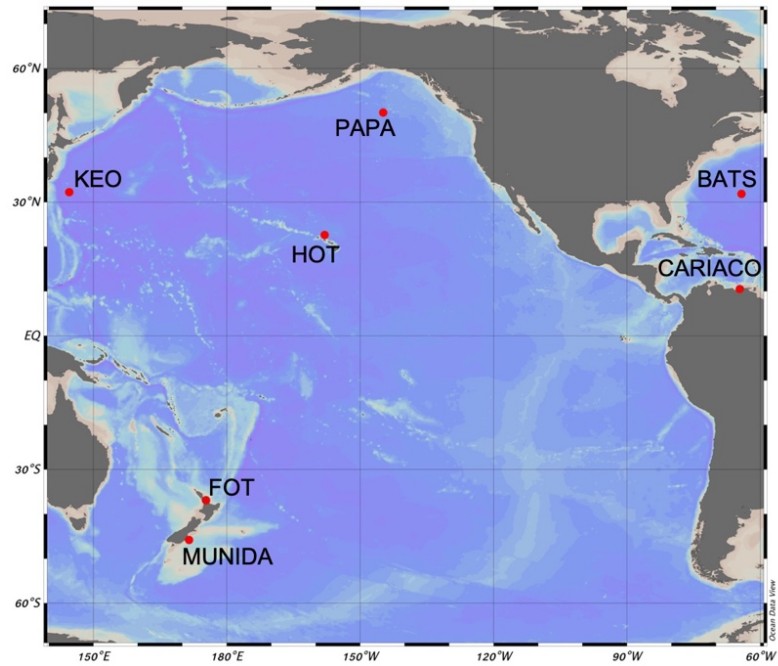


**Figure 1. Location map of seven ocean carbon time series sites utilized for estimating DIC using an empirical multiple linear regression model and other empirical and statistical approaches for imputing carbonate time series, including Bermuda Atlantic Time-series (BATS), Carbon Retention In A Colored Ocean (CARIACO), Firth of Thames (FOT), Hawaiian Ocean Time-series (HOT), Kuroshio Extension Observatory (KEO), Munida Time-series (Munida), and Ocean Site Papa (Papa). See Table 1 for**

**additional information about each sampling site.**

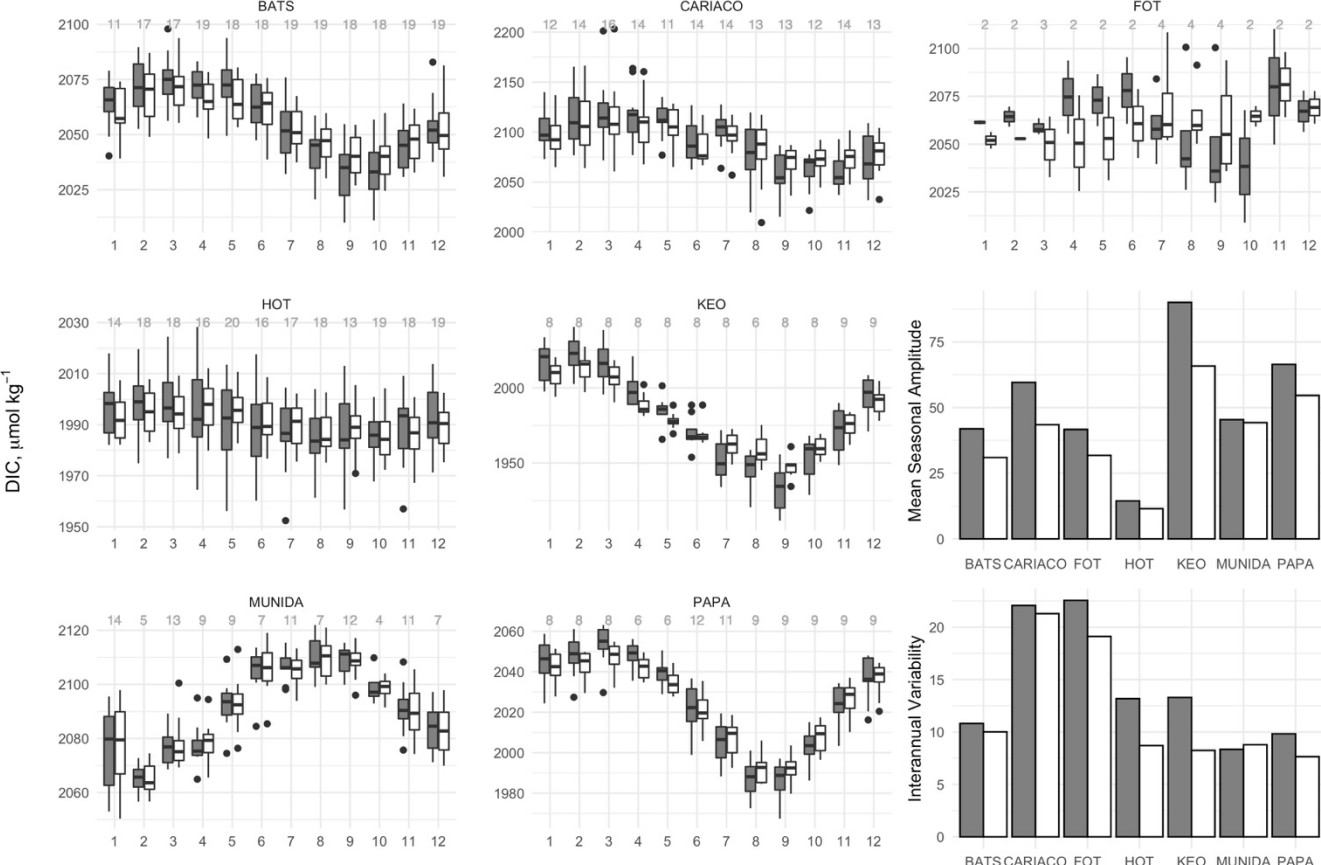

**Figure 2. Box and whisker plots of monthly mean concentrations of DIC (gray) and salinity normalized nDIC (white) in the mixed layer at each site, and bar plots showing the seasonal amplitude and interannual variability of DIC (gray) and nDIC (white). Box and whisker plots are composed of the median (solid line), lower and upper quartiles (box), the minimum and maximum values beyond the 25th and 75th quantile but < 1.5 interquartile range (whiskers) and values > 1.5 interquartile range (dots). Values above each box and whisker marker indicate the number of observations per month within the time series.**

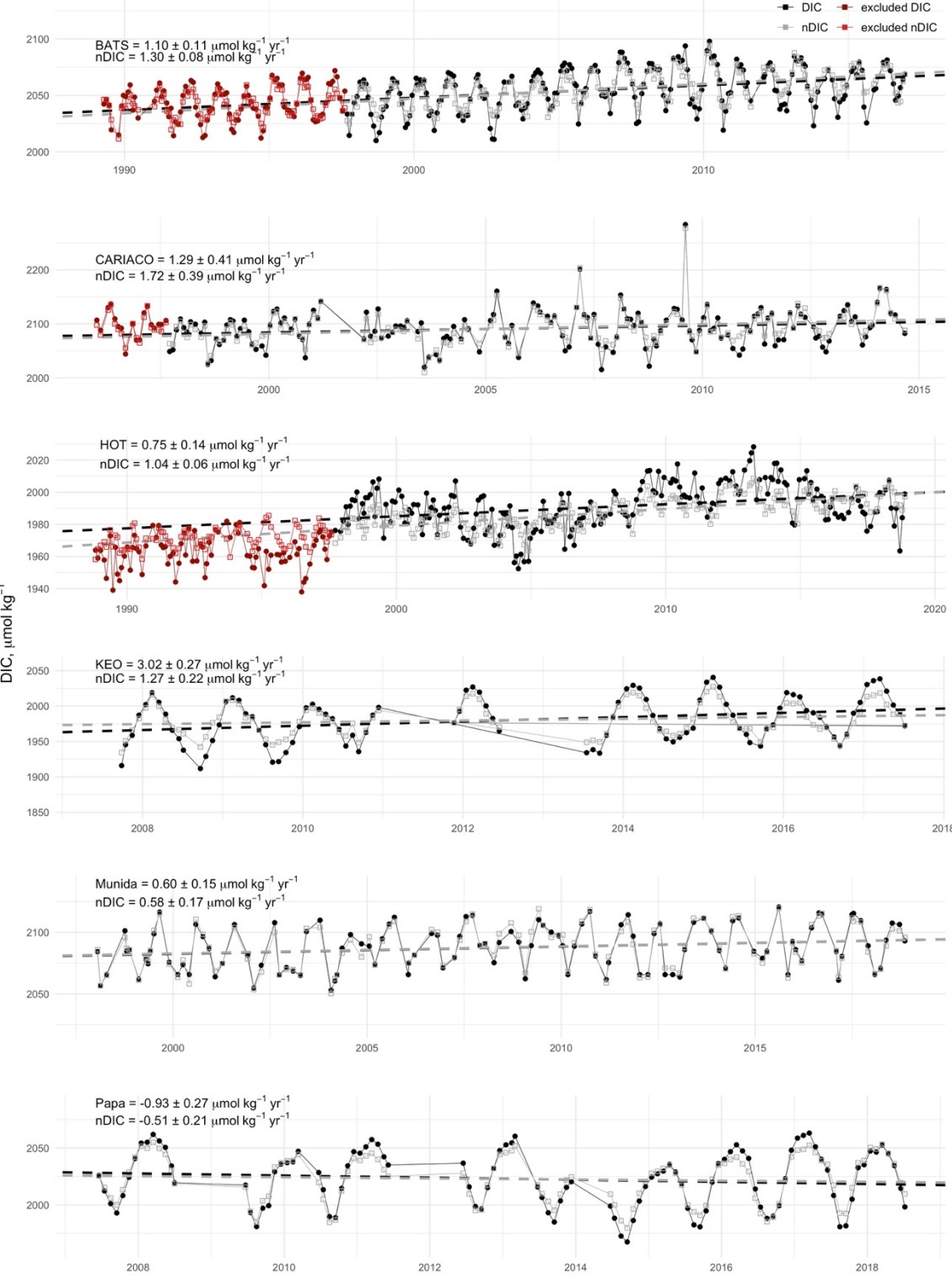

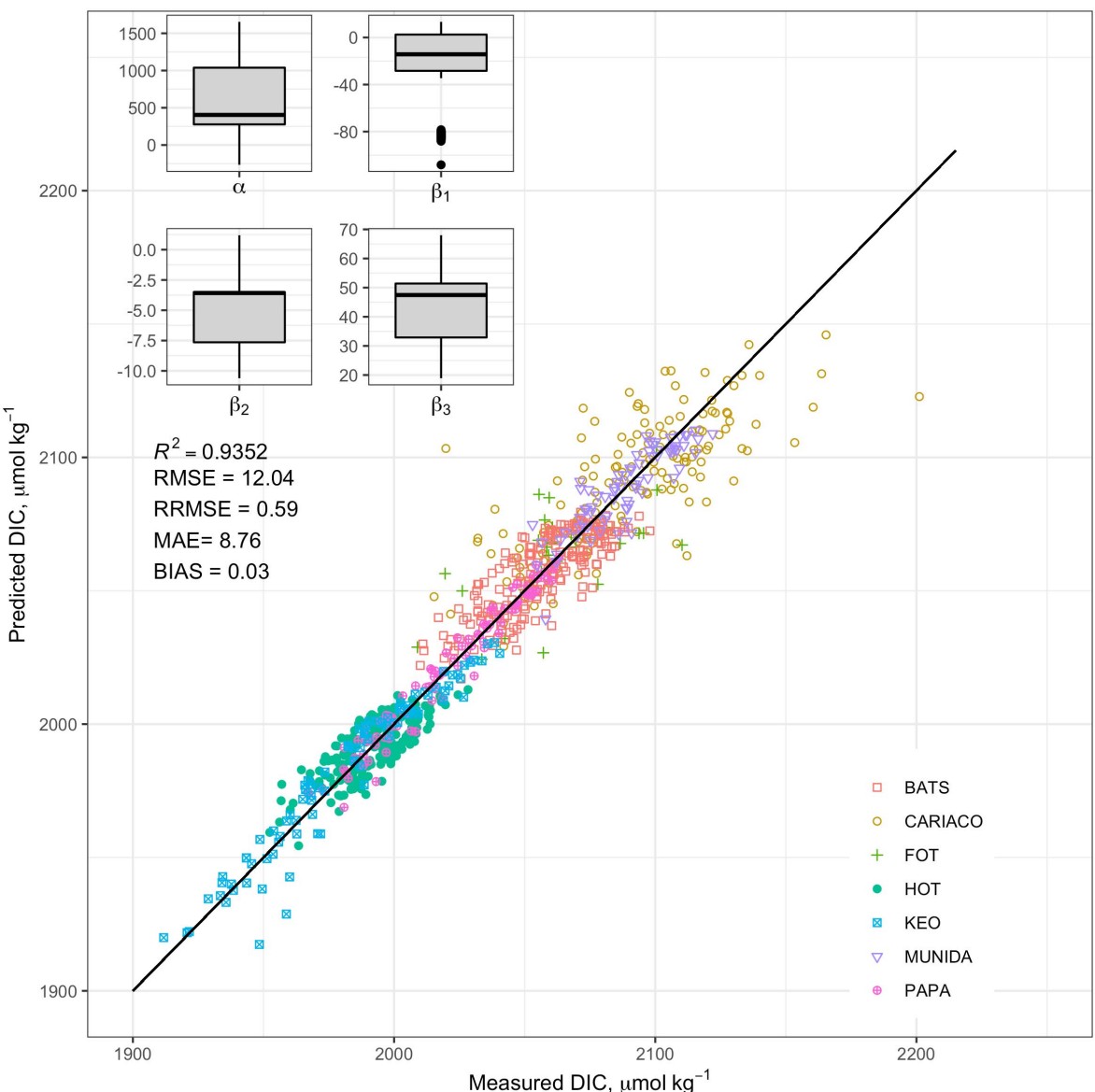

**Figure 3, Time series of DIC (black) and salinity normalized nDIC (grey) for each of the long-term data sets used to assess the impacts of gap-filling on the seasonal and interannual variability and long-term trends. Trends in seasonally detrended DIC with uncertainty are given for each site followed by the trend in nDIC below each value, and are shown as the corresponding dashed lines for each time series. Note that time series BATS, CARIACO and HOT were truncated to Sept. 1997, coincident with remotely sensed chlorophyll records and the data shown in red were excluded from analyses in this study.**

**Figure 4. Composite of predicted and measured DIC using a multiple linear regression model based on measured temperature, salinity and remotely sensed chlorophyll pooled from test sites: Bermuda Atlantic Time-series Study (BATS); Carbon Retention In A Colored Ocean (CARIACO); Firth of Thames (FOT); Hawaiian Ocean Time-series (HOT); Kuroshio Extension Observatory (KEO); Munida Time-series (Munida); Ocean Site Papa (Papa). Box and whisker plots for predictor variable coefficients $\alpha$, $\beta_1$ $\beta_2$ and $\beta_3$ are composed of the median (solid line), lower and upper quartiles (box), the minimum and maximum values beyond the 25th and 75th quantile but < 1.5 interquartile range (whiskers) and values > 1.5 interquartile range (dots).**




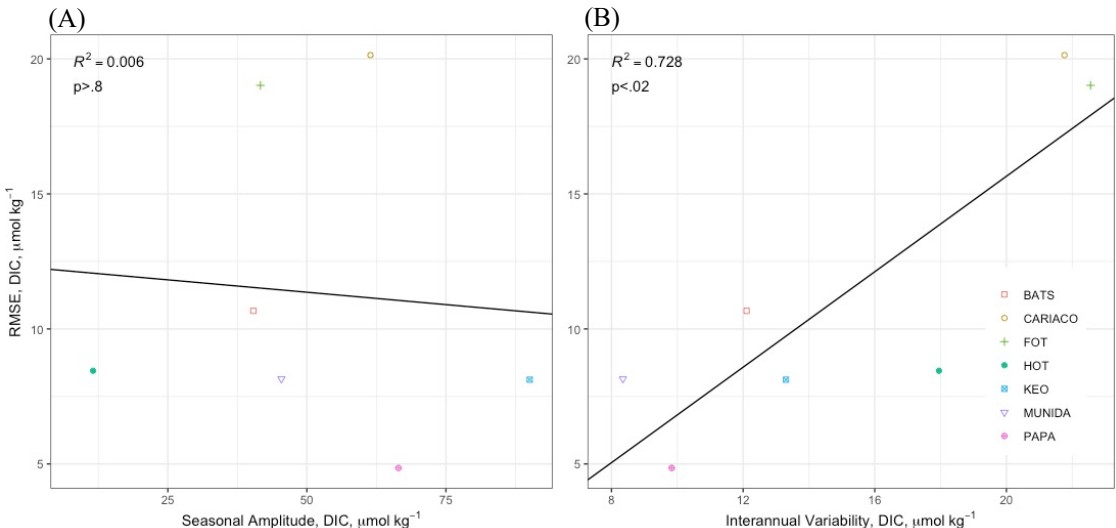

**Figure 5. Correlations between RMSE and (A) seasonal amplitude and (B) interannual variability across sites**

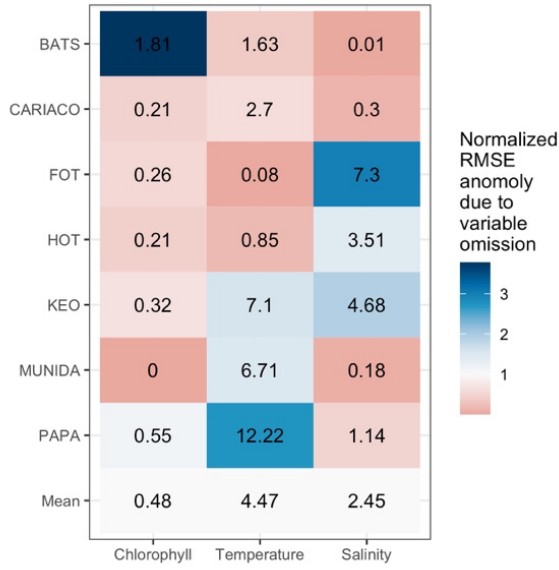


**Figure 6. Tile plot showing the change in RMSE per site due to the selective omission of input variables and refitting of the MLR. Tiles are colored to normalized error anomalies for visualization of relative differences, while RMSE anomalies are given in each tile for the effect of omitting the predictor variable at each site.**

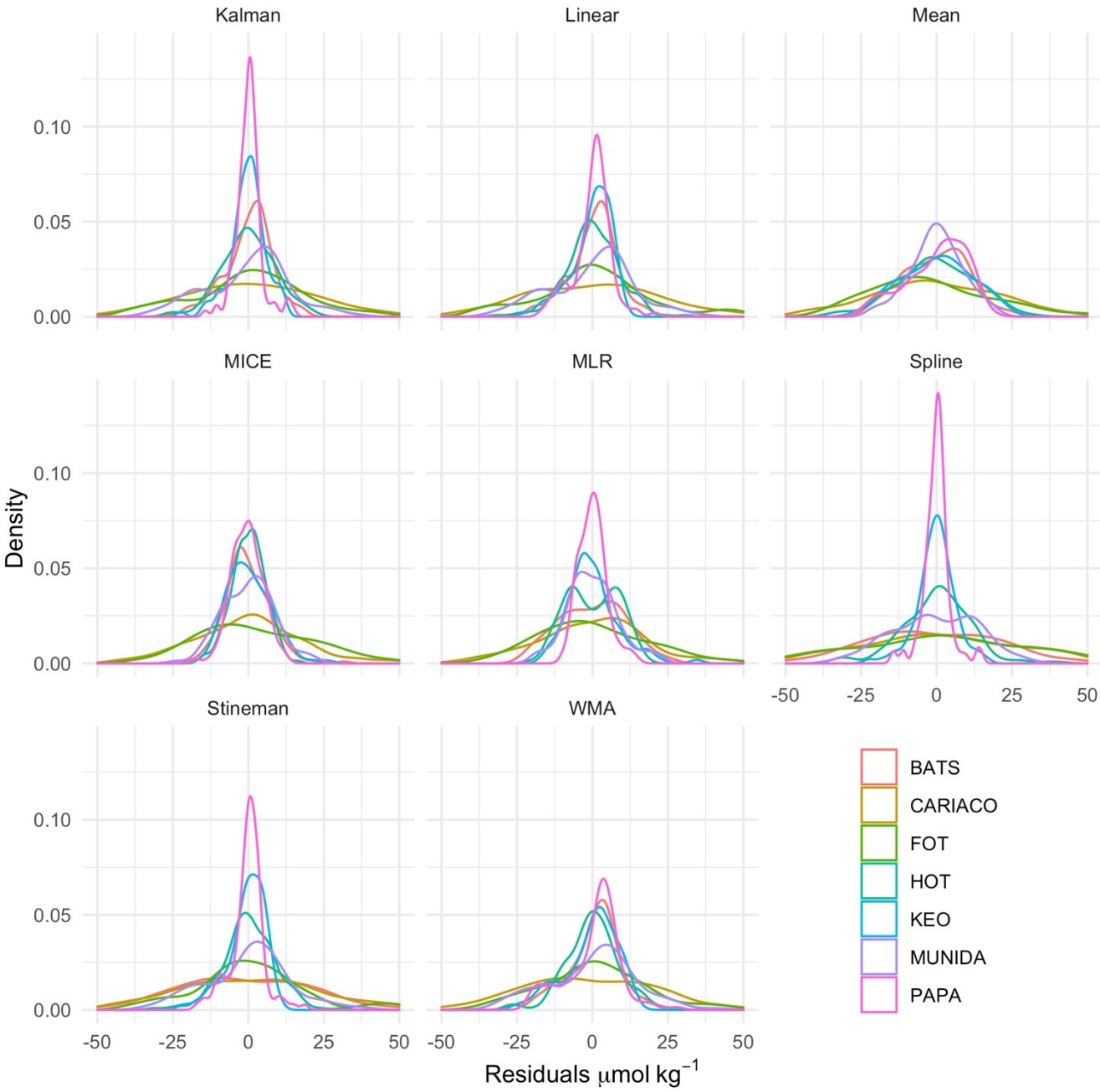

Figure 7. Kernel density curves of the DIC residuals between gap-filled and observed time series for each imputation model using Leave One Out Cross Validation, for all observations after Aug 1997 coinciding with availability of remotely sensed chlorophyll data. Density curves are scaled so area under the curve equals one. Plots show the probability distribution of the residuals for each model. Skewness and modalities away from 0 indicate biasing.

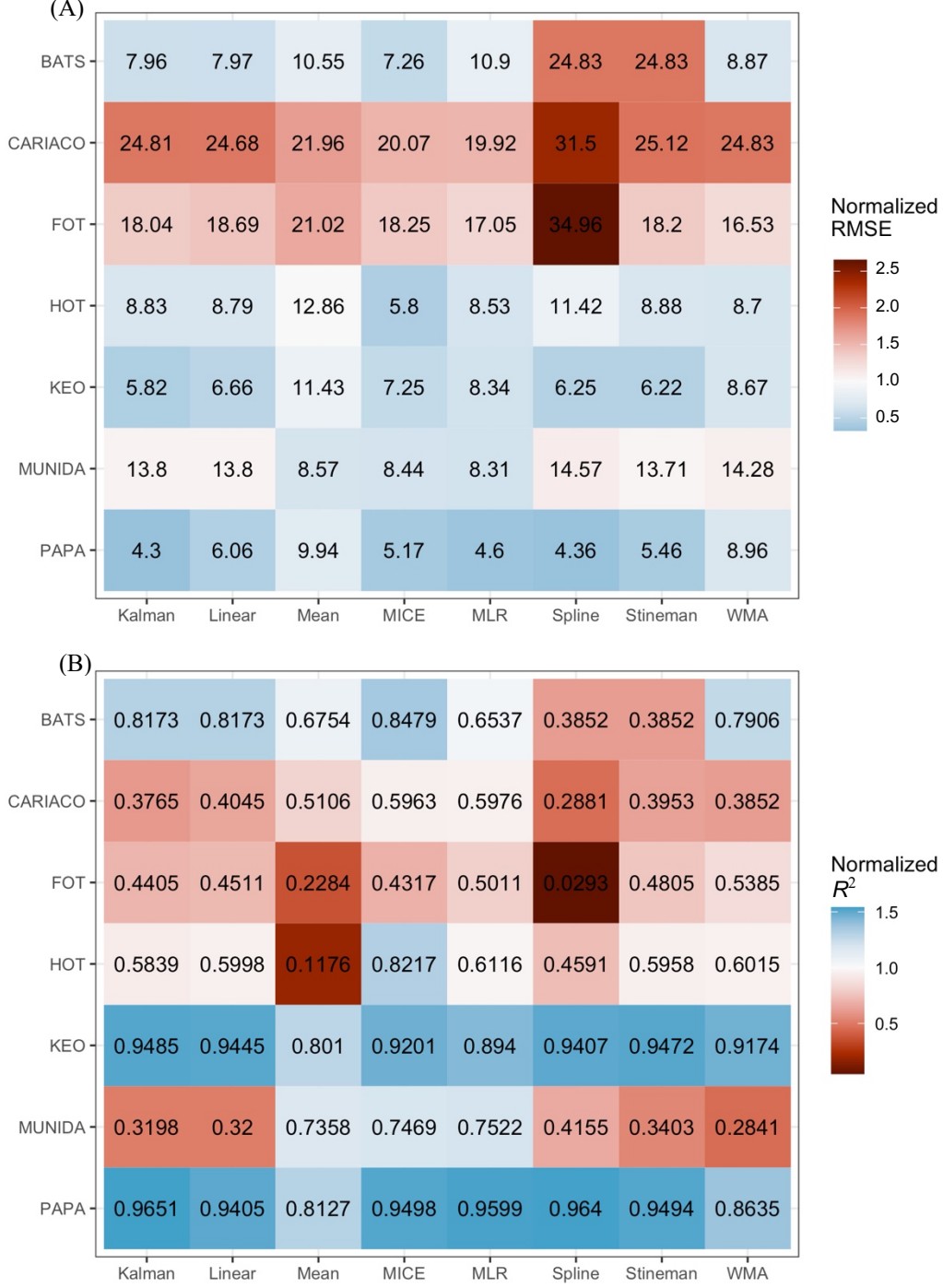


**Figure 8.** Tile plots showing **(A)** the RMSE (black text in tiles) for each cross validated imputation methods at each site. Tiles are colored according to RMSE normalized to the mean value across all methods and sites; and **(B)** the same format but for the squared correlation coefficient. Note errors at or below average performance do not equate to correlation that are average or better, e.g. Munida and HOT.

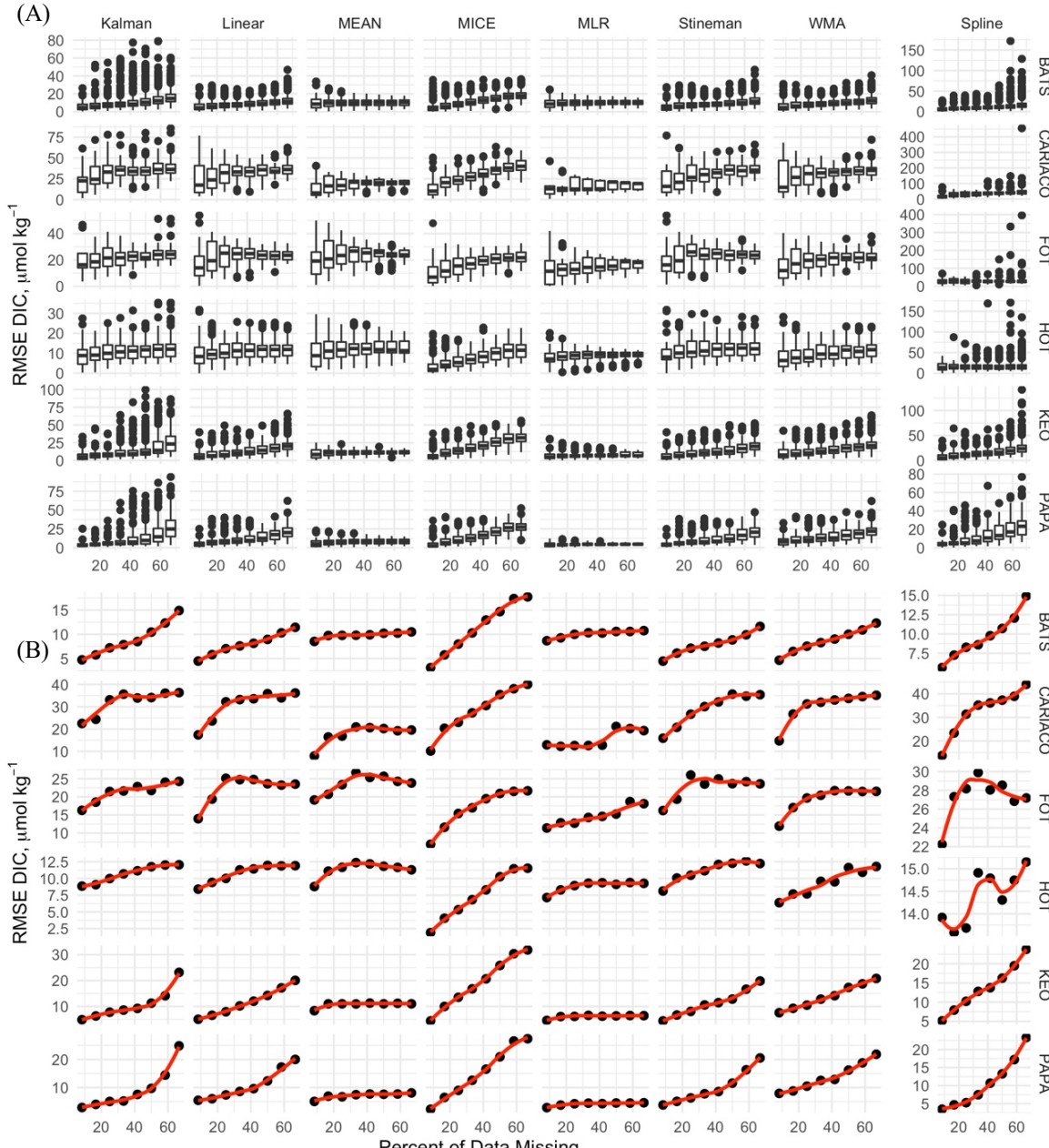


**Figure 9. (A)** Boxplots of RMSE for each gap assessment corresponding to 8.33%, 16.67%, 25%, 33.33%, 41.67%, 50%, 58.33% and 66.67% data missing rates. Box and whisker plots are composed of the median (solid line), lower and upper quartiles (box), the minimum and maximum values beyond the 25th and 75th quantile but < 1.5 interquartile range (whiskers) and values > 1.5 interquartile range (dots). Points above box and whiskers indicate the distribution of outliers for each model. **(B)** Loess fit (red line)
of the median error for each gap assessment, indicating the sensitivity of the model to increasing data loss. Scales adjusted per site.

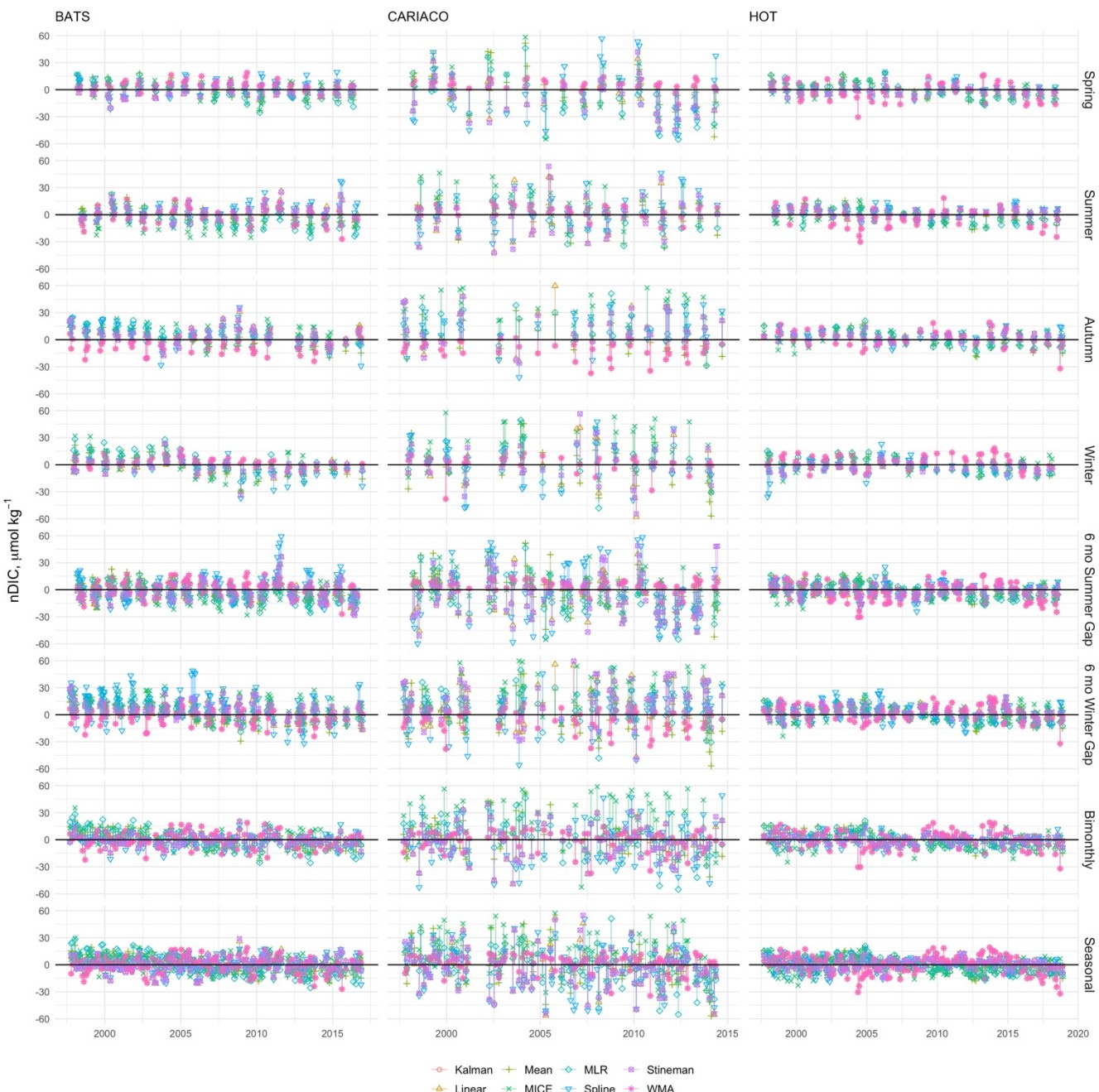

**Figure 10A. Residuals between imputed and observed nDIC from BATS, CARIACO and HOT. Observations were selectively removed using eight gap filters: 3-month sequential seasonal filters for Spring, Summer, Autumn, and Winter; 6-month sequential gaps centered on summer and winter; and bimonthly (odd months) and seasonal (1 max, 1 min. and 2 transition samples) sampling regimes and gaps were filled using Kalman filter with a state space model, linear interpolation, mean imputation, empirical multiple linear regression (MLR), multiple imputation by chained equations (MICE), spline interpolation, Stineman interpolation and exponential weighted moving average (WMA).**


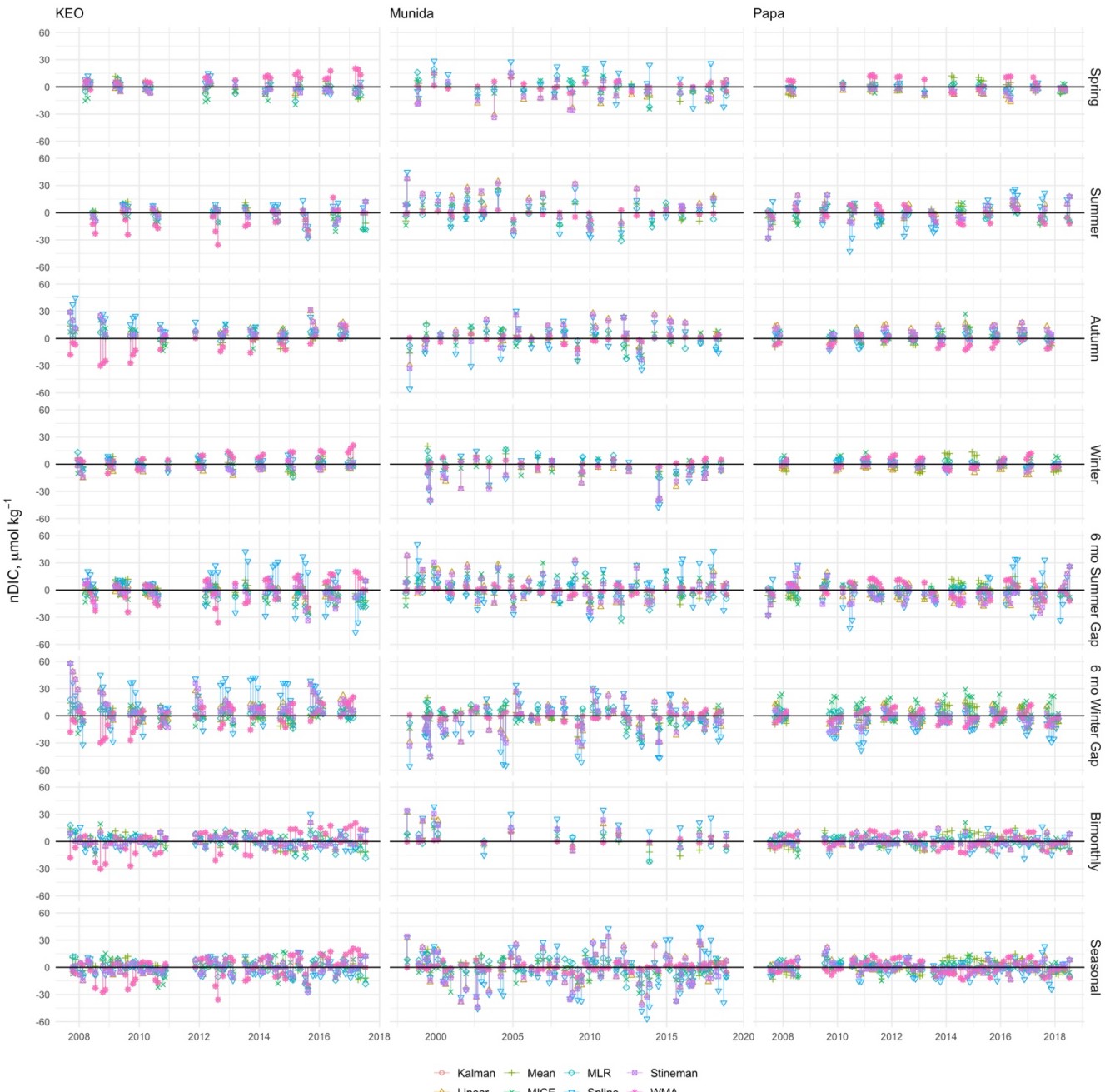

Figure 10B. Residuals between imputed and observed nDIC from KEO, Munida and Papa. Observations were selectively removed using eight gap filters: 3-month sequential seasonal filters for Spring, Summer, Autumn, and Winter; 6-month sequential gaps centered on summer and winter; and bimonthly (odd months) and seasonal (1 max, 1 min. and 2 transition samples) sampling regimes and gaps were filled using Kalman filter with a state space model, linear interpolation, mean imputation, empirical multiple linear regression (MLR), multiple imputation by chained equations (MICE), spline interpolation, Stineman interpolation and exponential weighted moving average (WMA).



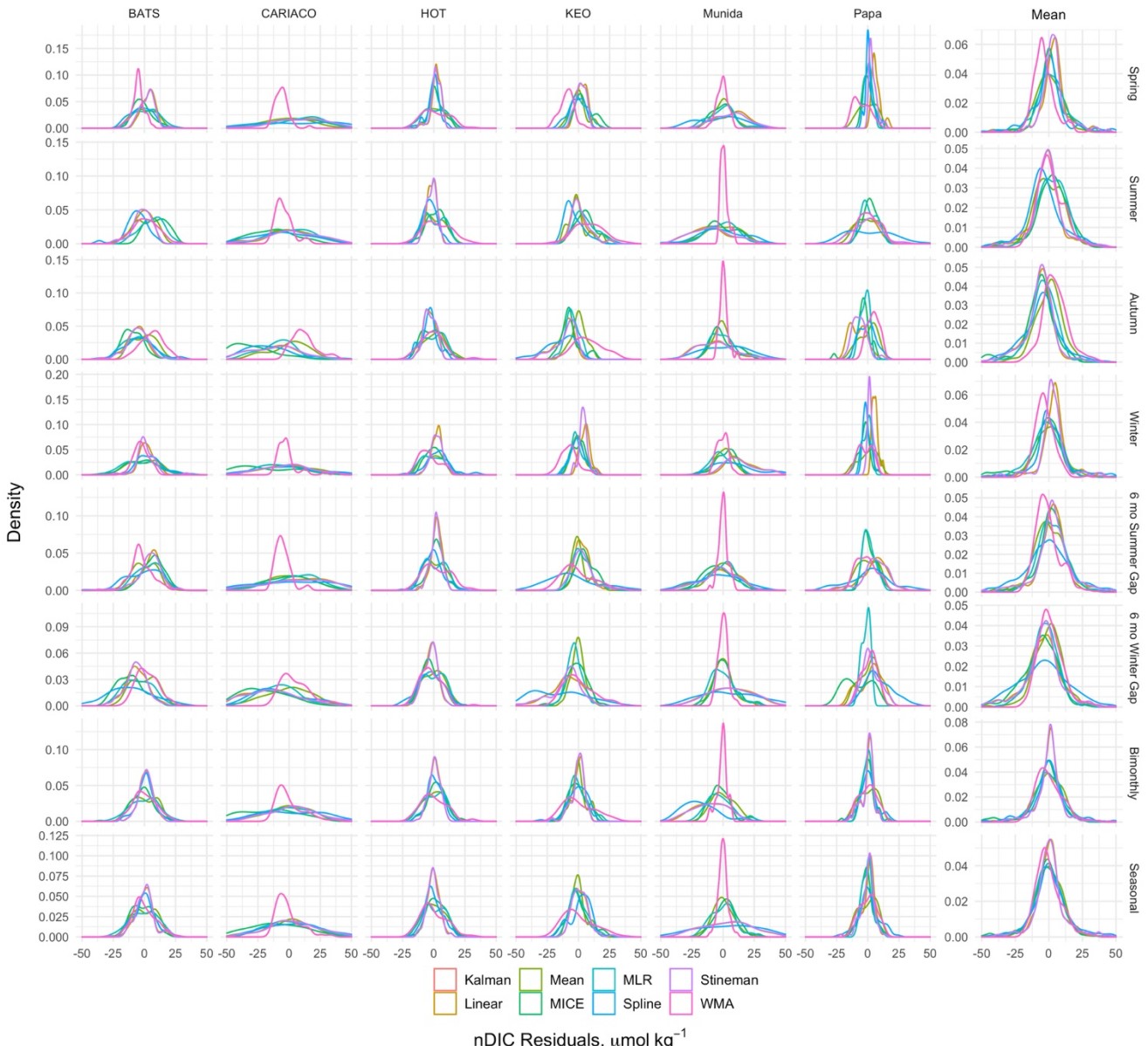

**Figure 11. Kernel density curves of the nDIC residuals between gap-filled and observed values for each site and synthetic gap filter tested (see also Fig. 10). Residuals pooled across sites are shown as the Mean column on the right-hand side. Density curves are scaled so area under the curve equals one. Plots show the probability distribution of the residuals. Skewness and modalities away from 0 indicate biasing.**

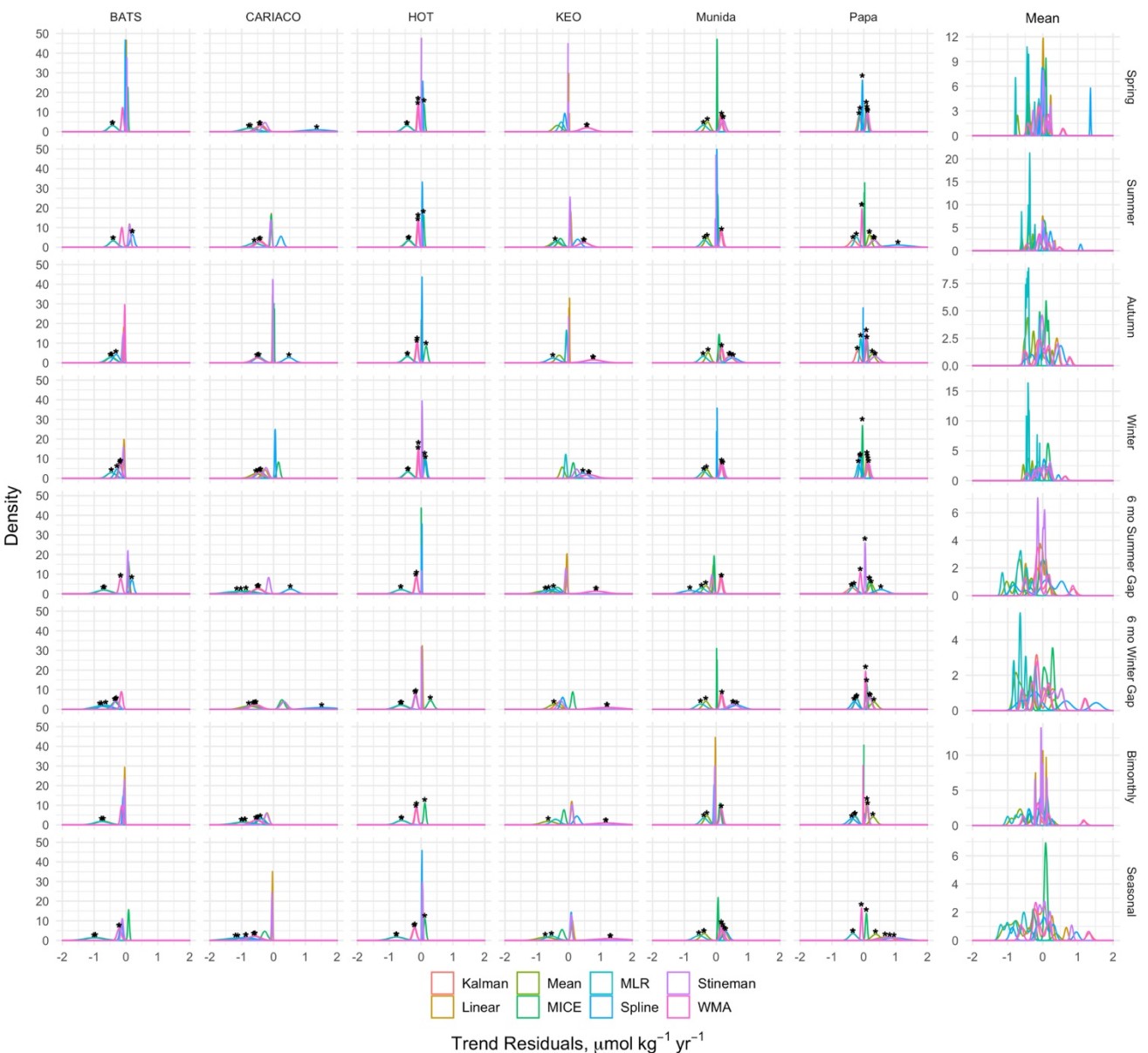

**Figure 12. Kernel density curves of the nDIC residuals between the trends calculated from observed and gap-filled time series for each site and synthetic gap filter tested (see also Figs 10-11). Residuals pooled across sites are shown as the Mean column on the right-hand side. Residuals that exceeded the uncertainty bounds for the observed trend are denoted with a black asterisk. Peaks to either side of 0 indicate positive or negative biasing in the imputation method resulting in changes in the apparent trend for the reconstructed time series.**

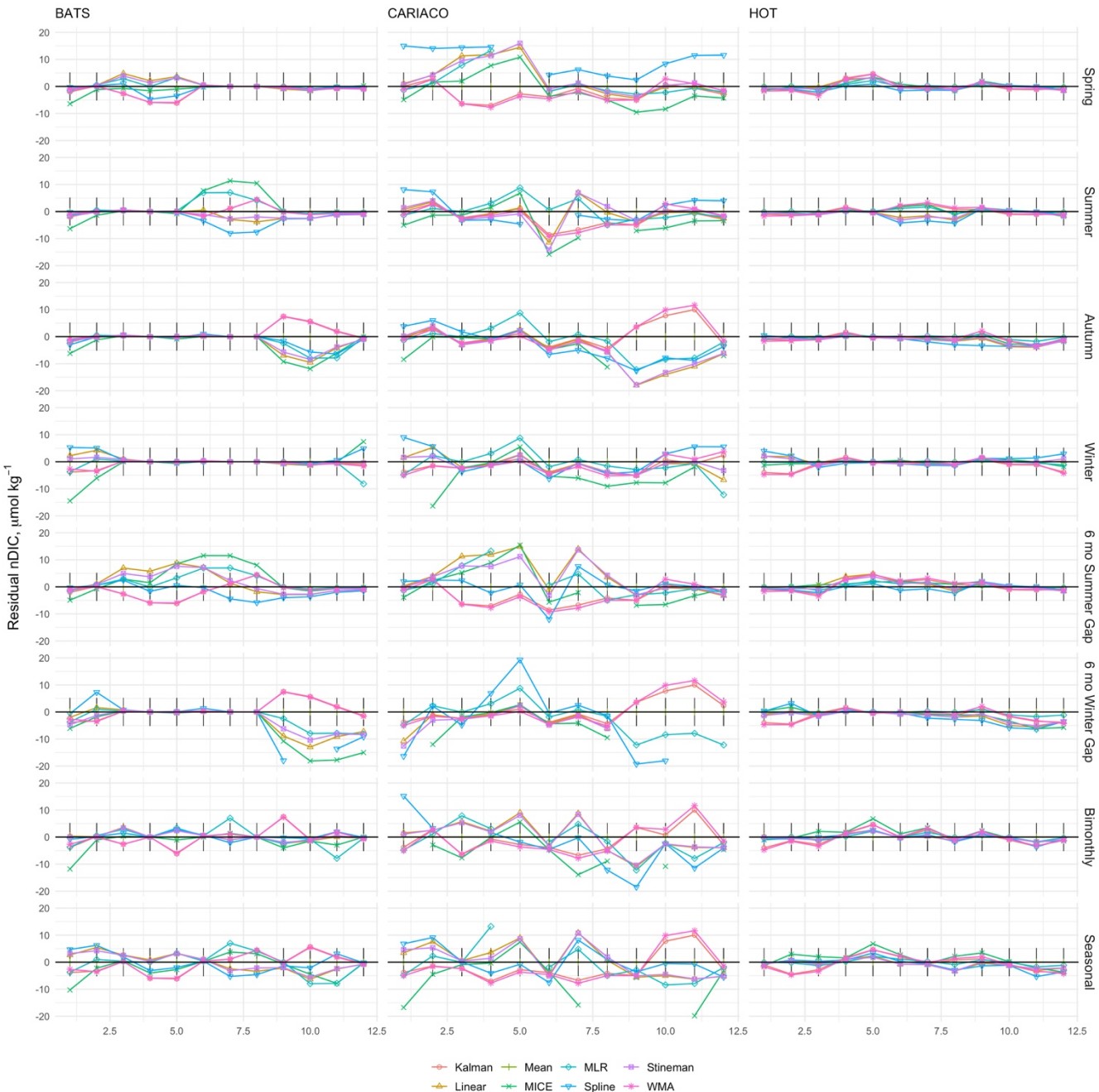

**Figure 13A. Residuals between climatological monthly means calculated from observed time series and reconstructed time series of nDIC from BATS, CARIACO, and HOT. Monthly means were calculated from the time series (individual residuals shown in Figs. 10A-B) values infilled by the eight imputation models. Black sticks represent the combined uncertainty for the observations at each site.**

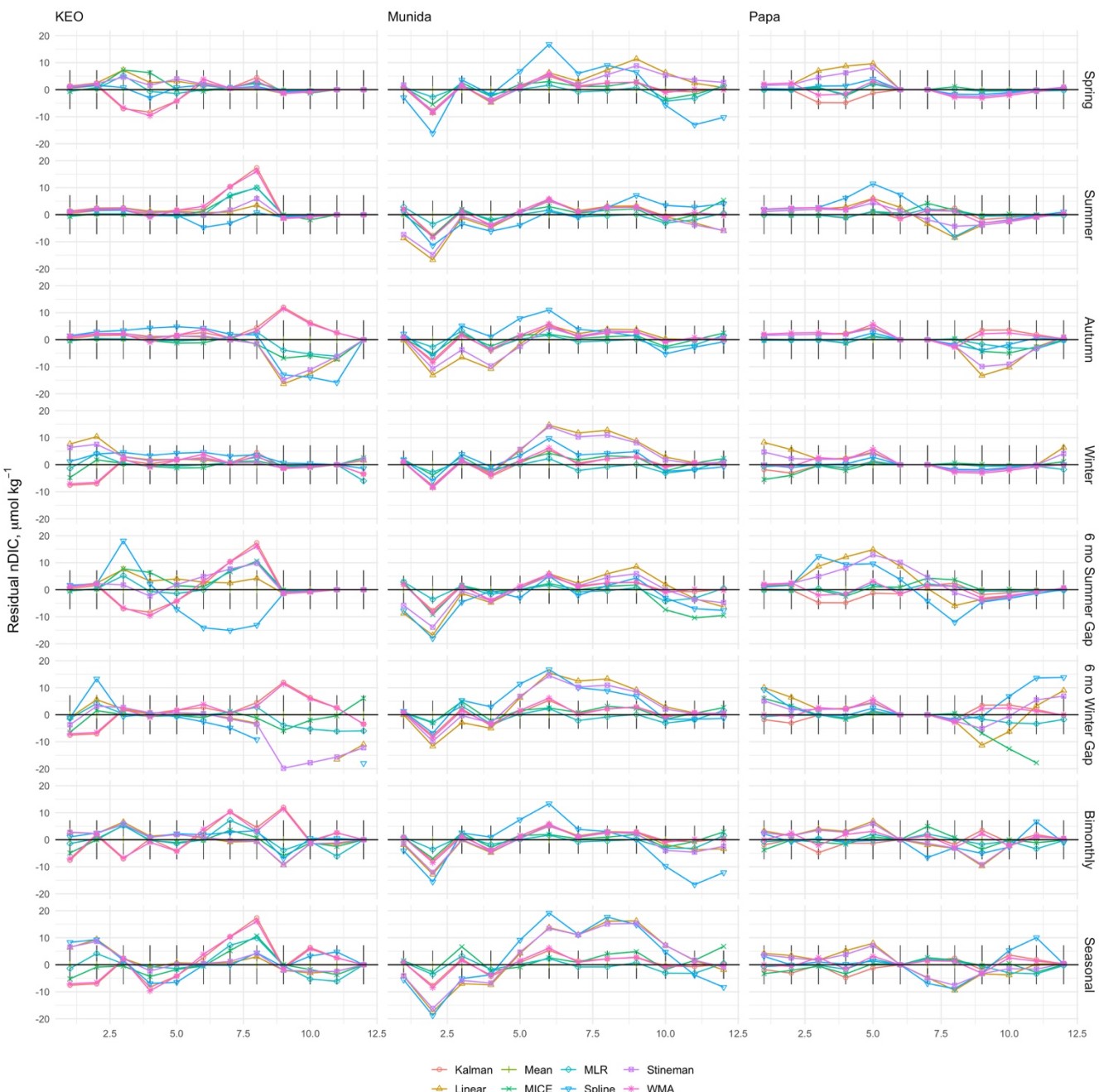

**Figure 14. Residuals between climatological monthly means calculated from observed time series and reconstructed time series of nDIC from KEO, Munida, and Papa. Monthly means were calculated from the time series (individual residuals shown in Figs. 10A-B) values infilled by the eight imputation models. Black sticks represent the combined uncertainty for the observations at each site.**

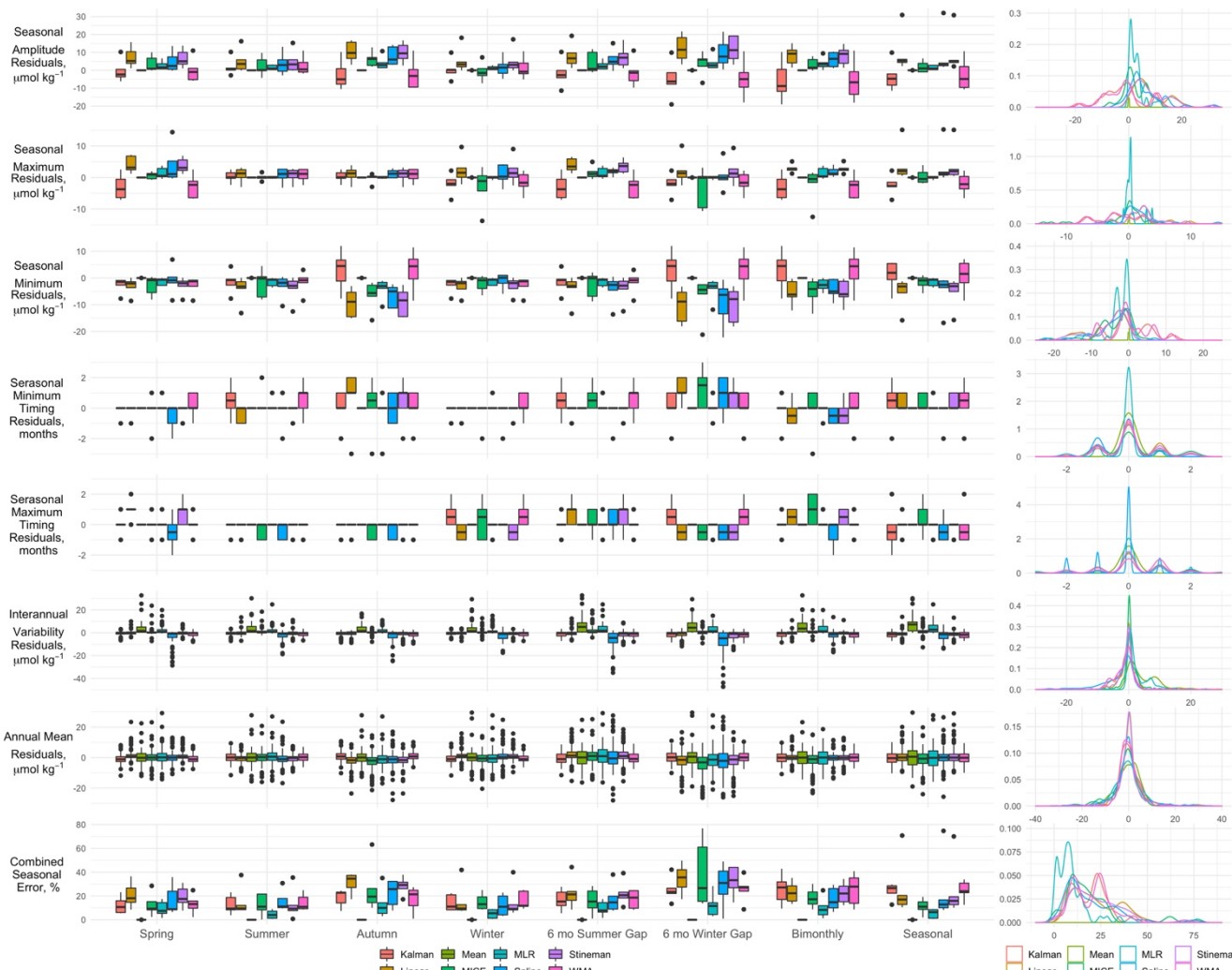

**Figure 14. Boxplots of the residuals between gap-filled and observed time series for: seasonal amplitude (difference between seasonal maximum and minimum); the seasonal maxima and minima, and their respective timing (the month when maxima and minima are observed); interannual variability (the standard deviation of monthly means); and the annual means. Combined Seasonal Error represents the combined absolute percent errors of the seasonal amplitude, maximum, minimum, and timing (see Eq.10). Box and whisker plots are composed of the median (solid line), lower and upper quartiles (box), the minimum and maximum values beyond the 25th and 75th quantile but < 1.5 interquartile range (whiskers) and values > 1.5 interquartile range (dots). The right-hand column shows the kernel density curves for each seasonal metric, pooled across all synthetic gap filters. Peaks in the density plots represents modes where mean errors for each model as associated with each gap filter.**


**Table 1. Information about each sampling site with ocean carbonate time series used in our analyses, including Bermuda Atlantic Time-series (BATS), Carbon Retention In A Coloded Ocean (CARIACO), Firth of Thames (FOT), Hawaiian Ocean Time-series (HOT), Kuroshio Extension Observatory (KEO), Munida Time-series (Munida), and Ocean Site Papa (Papa). DIC = dissolved inorganic carbon. TA = total alkalinity. $pCO_2$ = partial pressure of carbon dioxide. pH = -log[$H^+$]. Gap rate based on expected sampling frequency.**


| Site Type | Time series Site | Sampling Region | Location | Time series Duration | Sampling Frequency | Gap Rate | Carbonate Measurements |
|---|---|---|---|---|---|---|---|
| Sampling Site | BATS | Sargasso Sea | 31.88°N, 64.26°W | 1983 - present | [1]monthly | 4% | DIC/TA |
| | HOT | North Pacific | 22.67°N, 158°W | 1988 - present | [2]monthly | 15% | TA/pH |
| | CARIACO | Cariaco Basin | 10.5°N, 64.67°W | 1995 - present | monthly | 16% | TA/pH |
| | MUNIDA | South Pacific | 45.8°S 171.5°E | 1998 - present | [3]bimonthly | 5% | $pCO_2$/TA |
| Mooring | PAPA | North Pacific | 50.13°N, 144.83°W | 2007 - present | 3 hours | 26% | pH/$pCO_2$ |
| | KEO | North Pacific | 32.25°N, 144.56°E | 2004 - present | 3 hours | 18% | pH/$pCO_2$ |
| | FOT | New Zealand Coast | 36.88°S, 175.32°E | 2015 - present | [*]15 minutes | 59% | pH |

Web addresses for site information and data access:
BATS: http://www.bios.edu/research/projects/bats/
HOT: https://hahana.soest.hawaii.edu/hot/
CARIACO: http://www.imars.usf.edu/cariaco
Munida: https://marinedata.niwa.co.nz/nzoa-on/
Papa: https://www.pmel.noaa.gov/ocs/Papa
KEO: https://www.pmel.noaa.gov/ocs/KEO

FOT: https://marinedata.niwa.co.nz/nzoa-on/
[*]Sampling began in 1998, mooring installed in 2015
[1]BATS sampling target is at least monthly
[2]HOT sampling target is approximately monthly
[3]Munida sampling is typically bimonthly, varying with conditions and additional coordinated voyages

**Table 2. Pearson correlation coefficients between DIC and chlorophyll, temperature and salinity in the surface layer across test sites. Asterisks indicate weak correlations (threshold = 0.3).**

| Site | Pearson Correlation Coefficient | | |
|------|------------|-------------|----------|
| | Chlorophyll | Temperature | Salinity |
| KEO | 0.49 | -0.91 | 0.87 |
| BATS | 0.48 | -0.73 | 0.65 |
| Papa | -0.34 | -0.97 | 0.73 |
| FOT | -0.22* | 0.24* | 0.74 |
| HOT | 0.1* | -0.51 | 0.74 |
| CARIACO | 0.53 | -0.77 | 0.58 |
| Munida | -0.37 | -0.87 | 0.32 |

**Table 3. Years with 12 monthly samples per site. *Actual sampling interval greater than monthly**

| Time-Series Site | Years With 12 Monthly Samples | N Years |
|------------------|-------------------------------|---------|
| BATS | 1998, 1999, 2000, 2001, 2004, 2005, 2007, 2008, 2012, 2013 | 10 |
| HOT | 1998, 2004, 2006 | 3 |
| CARIACO | 2008 | 1 |
| Munida | NA* | 0 |
| Papa | 2015, 2016, 2017 | 3 |
| KEO | 2009, 2010, 2014, 2015, 2016 | 5 |
| FOT | 2016 | 1 |

**Table 4. Results of cross validated MLR model for estimating DIC at each individual site, and at grouped oceanic (BATS, HOT, KEO, Munida, Papa) and coastal (FOT, CARIACO) sites, including the mean and standard deviation of each coefficient for $N$ LOOCV iterations.**

| Site | RMSE | RRMSE | $R^2$ | MAE | BIAS | $N$ | $\alpha$ | $\beta_1$ | $\beta_2$ | $\beta_3$ |
|---|---|---|---|---|---|---|---|---|---|---|
| BATS | 10.67 | 0.52 | 0.6611 | 8.93 | 0.017 | 208 | 401.65±13.75 | -13.48±1.56 | -3.53±0.03 | 47.53±0.36 |
| CARIACO | 20.14 | 0.96 | 0.5861 | 14.94 | 0.015 | 153 | 1446.46±40.07 | 2.50±0.10 | -10.16±0.12 | 24.37±1.02 |
| FOT | 19.02 | 0.92 | 0.3958 | 15.13 | 0.099 | 28 | 718.32±47.59 | 8.30±2.53 | 0.47±0.35 | 37.93±1.26 |
| HOT | 8.45 | 0.42 | 0.6178 | 7.40 | 0.029 | 204 | 276.44±9.51 | -82.88±2.25 | -3.47±0.04 | 51.44±0.26 |
| KEO | 8.12 | 0.41 | 0.9330 | 6.12 | 0.061 | 90 | -208.45±16.79 | -27.85±1.01 | -4.61±0.03 | 66.36±0.48 |
| Munida | 8.15 | 0.39 | 0.7564 | 6.48 | 0.029 | 109 | 1069.11±65.27 | 4.77±1.05 | -7.69±0.08 | 32.00±1.89 |
| Papa | 4.85 | 0.24 | 0.9631 | 3.74 | 0.035 | 94 | 799.13±17.96 | -16.47±0.52 | -6.55±0.02 | 39.82±0.55 |
| Oceanic | 8.75 | 0.43 | 0.9567 | 7.09 | 0.030 | 671 | 412.04±356.85 | -34.86±32.81 | -4.54±1.53 | 48.5±9.35 |
| Coastal | 19.97 | 0.95 | 0.6078 | 14.97 | 0.028 | 181 | 1333.82±267.23 | 3.40±2.32 | -8.52±3.86 | 26.47±5.03 |

**Table 5. Mean model results for selective omission of input variables.**

| Variable Omitted | RMSE | RRMSE | $R^2$ | MAE | BIAS |
|---|---|---|---|---|---|
| none | 12.044 | 0.591 | 0.9352 | 8.764 | 0.030 |
| chlorophyll | 12.106 | 0.594 | 0.9345 | 8.849 | 0.005 |
| temperature | 15.526 | 0.762 | 0.8923 | 11.871 | 0.013 |
| salinity | 13.998 | 0.687 | 0.9124 | 10.285 | 0.022 |

**Table 6. Performance metrics for cross validated imputation models across all sites.**

| Model | RMSE | RRMSE | $R^2$ | MAE | BIAS |
|---|---|---|---|---|---|
| Kalman | 13.22 | 0.65 | 0.9230 | 8.74 | -0.03 |
| Linear | 13.34 | 0.65 | 0.9218 | 9.00 | -0.02 |

| | | | | | |
|---|---|---|---|---|---|
| Mean | 13.91 | 0.68 | 0.9149 | 10.51 | 0.00 |
| MICE | 10.78 | 0.53 | 0.9489 | 7.17 | 0.07 |
| MLR | 11.75 | 0.58 | 0.9392 | 8.57 | 0.03 |
| Spline | 19.89 | 0.97 | 0.8672 | 13.29 | -0.43 |
| Stineman | 16.91 | 0.83 | 0.9013 | 11.53 | -0.28 |
| WMA | 13.79 | 0.68 | 0.9163 | 9.69 | -0.09 |

1010

**Table 7. Percent of missing data associated with synthetic gap filters applied to each time series, the number observations, total months, and percent missing observations based on a monthly frequency for the time series duration tested.**

| Site | Spring | Summer | Autumn | Winter | 6-month Summer Gap | 6-month Winter Gap | Bimonthly | Seasonal | n Obs. | Months | % of Missing Obs. |
|---|---|---|---|---|---|---|---|---|---|---|---|
| BATS | 32% | 33% | 33% | 29% | 56% | 53% | 53% | 71% | 212 | 233 | 9% |
| CARIACO | 42% | 42% | 41% | 41% | 62% | 60% | 61% | 75% | 160 | 206 | 22% |
| HOT | 41% | 39% | 39% | 39% | 61% | 59% | 59% | 74% | 206 | 256 | 20% |
| KEO | 33% | 32% | 35% | 35% | 53% | 59% | 57% | 71% | 105 | 119 | 12% |
| Munida | 67% | 67% | 69% | 67% | 78% | 79% | 63% | 85% | 109 | 252 | 57% |
| Papa | 30% | 37% | 34% | 34% | 55% | 57% | 55% | 70% | 118 | 134 | 12% |

1015