# Peer review of "An Empirical MLR for Estimating Surface Layer DIC and a Comparative Assessment to Other Gap-filling Techniques for Ocean Carbon Time Series"

_Biogeosciences, 2021_

## Author Comment (AC1)

| Index | Line (initial) | Comment | Reviewer | Response (revision or comment) |
|---|---|---|---|---|
| 1 | Section 2.6 | For the error propagation described in section 2.6, what is the reasoning for using CO2 flux? Using CO2 flux introduces several other potential biases and errors to the assessment: | 1 | The initial reasoning for inlcuding the CO2 flux was to show implications of error propogation for various imputations during a common use case for DIC time series. However, both reviewers have raised similar concern about the introduction of multiple sources of error when determining CO2 flux. The combined uncertainty for the CO2 flux was initially determined by a Monte Carlo method (n=1000, which was not significantly different than n=10000) and then only the values of imputed DIC and their associeted uncertainty were varied as inputs into the calculation. In this way we attributed the percent difference between imputed time series and observed time series to be related only to the gap-filling method because no other input was varied. Similarly the uncertnainty of the CO2 flux was determined via this MCM for each method. That said, we understand the concern about multiple sources of error and recognize that this application detracts from the results of just gap-filling the DIC time series. We will remove the CO2 flux aspect of paper and add a focus on long term trend assessment in its place. This will be more consistent with the intentions of the work, enhance the focus of the paper and address mulitple comments from both reviewers. |
| 2 | | · uncertainty in air pCO2, | 1 | removing CO2 flux aspect |
| 3 | | · major bias and errors of NCEP winds (see: https://doi.org/10.5194/bg-15-1701-2018, https://doi.org/10.1029/2018GB006047, https://doi.org/10.1002/2017GL073814) | 1 | removing CO2 flux aspect |
| 4 | | · uncertainty in the gas transfer velocity coefficient (resulting in total uncertainty in CO2 flux of ~20%), and | 1 | removing CO2 flux aspect |
| 5 | | · uncertainty (~5%) introduced in the calculation of sw pCO2 from DIC and TA. | 1 | removing CO2 flux aspect |
| 6 | | How will those biases and errors complicate your assessment of gap filling error propagation? The relative uncertainty for CO2 flux at BATS is reported in line 361 as 3.5%. What does this uncertainty take into account? Not items 1 – 4 above, as this value would be much higher. These issues should be addressed in the error propagation, or another parameter should be used for this assessment. | 1 | removing CO2 flux aspect |
| 7 | NA | Data used in this study need to be cited properly, which is incredibly important to the programs supporting these time series measurements. Those data should be cited in the methods and/or funders noted in the acknowledgements, depending on what each time series program recommends, not recorded as web addresses in the notes of Table 2. For the moorings, if you are accessing original data files via NCEI, those citations can be found at https://doi.org/10.3334/cdiac/otg.tsm_papa_145w_50n for Papa and https://doi.org/10.3334/cdiac/otg.tsm_keo_145e_32n for KEO. If you are accessing the mooring data from the synthesis product, the citation can be found at https://doi.org/10.7289/V5DB8043. I am not as familiar with the citation requirements of all the ship-based time series, but with a quick search I found this data citation request for HOTS, for example: https://hahana.soest.hawaii.edu/hot/dataaccess.html | 1 | This was a gross oversight on our part. While the sources for data sets were listed in table 1, they were not properly cited as noted. We will cite these as required. |

| | | | | |
|---|---|---|---|---|
| 8 | NA | Finally, it may be out of the scope to include additional analyses in this paper, but it would be worthwhile discussing future work that can build off these results. For example, what satellite-based products are best suited for the MLR approach? Are there any that can span open ocean and coastal environments, so gap filling methods can be applied consistently across all global ocean and coastal time series? Also, it would be useful to study whether there are discrepancies in calculated trends when using these different gap filling methods (at least the most successful methods) or no gap filling methods at all. Both of these analyses seem like they could have been included in this paper, but I could also understand if those are the next assessments planned using the most promising empirical gap filling methods resulting from this work. | 1 | These are excellent points some of which we can address in the revision. Firstly, we will include an assessment of impacts on trends in place of the CO2 flux. Secondly, we have already separately performed a cross shelf assessment of the MLR performance using data from the Munida transect and we can included this appliaction. These aspect taken together will also help address Reviewer 2's comment about focusing the scope of the paper more on presenting this MLR method and comparing it to other gap-filling methods, rather than an extensive comparitive assessment of techniques since we have only selected a few methods from a very large number of possiblities. |
| 9 | 31 | Use the most recent version of the Global Carbon Budget: https://doi.org/10.5194/essd-12-3269-2020 | 1 | updated reference |
| 10 | 89 | Define DT | 1 | This was formatting issue only; this is really delta T, but the Greek symbol was lost at somepoint in the upload. |
| 11 | 89 | State the sites that did not measure DIC directly as in line 87 for discrete sampling sites | 1 | done |
| 12 | 90 | What measured parameters are being used to calculate DIC from the moored data? Measured pCO2 and pH? The measured pCO2 and pH pair has several issues, most importantly in this application is the issue brought up below for line 118, in that data return from pH sensors tend to be poor and data gaps will usually fall at the same time each year. Data return from the pCO2 systems are much better, and you will avoid much of the repeated seasonal gaps if you used established salinity-alkalinity relationships (in the Fassbender references) for those open ocean locations paired with measured pCO2 as discussed in https://doi.org/10.5194/bg-13-5065-2016. This will increase N Years in Table 3 for Papa and KEO | 1 | We calculated DIC for both KEO and Papa from measured pCO2 and a calculated total alkalinity using the published salinity algorithms from Fassbender et al 2016 and 2017, as you suggest. We will make sure this is more clearly communicated in the methods section for these sites as well as all other sites that do not mearusre DIC directly. |
| 13 | 96-99 | It would be useful to present more information (figure or some statistics like mean diff and standard deviation) about how MODIS and VIIRS compare at this particular site so it is more clear why VIIRS was chosen. | 1 | We will address this in revision |
| 14 | 118 | "Missing at random" is not a good assumption for many of the moored time series, especially the open ocean sites which tend to be serviced around the same time every year. Sensor failures are more likely late in the deployment, which can be around the same time every year just before servicing. That should be acknowledged here. | 1 | This has been addressed |
| 15 | 202 | BATS is a different latitude than Mauna Loa, and therefore, has different annual mean and seasonality of air xCO2. xCO2 air from same latitude of BATS should be used from one of these products: | 1 | removing CO2 flux aspect |
| 16 | | https://www.esrl.noaa.gov/gmd/ccgg/obspack/our_products.php | 1 | removing CO2 flux aspect |
| 17 | | https://www.esrl.noaa.gov/gmd/ccgg/carbontracker/ | 1 | removing CO2 flux aspect |
| 18 | 344 | What about: https://doi.org/10.1002/lom3.10232 and https://doi.org/10.3389/fmars.2017.00128? | 1 | Will review these references during revision |
| 19 | 358 | You should note that the studies cited here do not use ocean DIC time series. Include information on what types of time series these are (soil flux and respiration, etc). | 1 | This has been addressed |
| 20 | 406-408 | Since trends were not considered in this paper, this statement may be a bit premature? | 1 | This will be address / revised according to shift from CO2 flux to trend assessment |
| 21 | 655 | What is the note with the "*" referring to? | 1 | This has been addressed |
| 22 | Fig 10 | Why aren't the models listed above the top panel? And spline should maybe be presented on the far right or left since it has a diff y axis for the 6 month gap? | 1 | This was a formatting typo in R, but this figure will be removed per the removal of the CO2 flux assessment |

| | | | | |
|---|---|---|---|---|
| 23 | Fig 12 | Consistent with earlier comments about error propagation for CO2 flux, these results showing higher uncertainty at higher outgassing and uptake values are consistent with increased uncertainty at higher wind speeds. This makes it difficult to understand what is a gap filling uncertainty vs uncertainty in other parameters that impact flux. | 1 | As stated above, this assessment was meant to illusrate the change related to gap-filling method only since all other input data were held constant during the MCM analyses.However, this will also be removed during revision |
| 24 | 66 | On Line 66 is stated "This study aims to identify the optimal gap-filling methods for carbonate time series by establishing which techniques perform with sufficiently low error and bias to assess seasonal and interannual variability of carbonate biogeochemistry and the biological and physical processes that determine it." The manuscript takes the approach that all gap-filling techniques have been explored and that MLR is recommended as the best performing. While the latter is certainly true of the methods compared, I feel it is not currently possible to say the former while one / a number of machine learning (and other) approaches are absent - these have recently been successfully applied in oceanographic research, and so the manuscript is not fulfilling its own aims by omitting them. Clearly it is not feasible to compare all available methodologies, so I would recommend that you either tone down the aims of the paper (by saying that you present a MLR method for DIC time-series data gap imputation and compare it to other common, computationally inexpensive methods) or a selection of additional methods are included e.g. median as well as mean, machine learning (i.e. neural network, regression trees, random forests that you already mention), curve fitting, exponential moving average, k-nearest neighbours etc. | 2 | This point is well-taken. Given that we have not here (and could not really) assessed all methods, we will shift the stated focus away from optimization of gap-filling and toward presenting the MLR and comparing it against other common approaches as suggested. |
| 25 | NA | When comparing methods a lot of focus is on the magnitude of the RMSE. I feel the reader would benefit from some consideration of the structure of the error e.g. are certain times of the year subject to greater uncertainties, do the models reproduce the timing of the seasonal cycle, and the magnitude of the peaks and troughs or are these far worse than those that vary around annual mean values? Equally, is the error of the preferred MLR technique actually normally distributed, as a lot of its power rests on this assumption. The manuscript would certainly benefit from greater examination of the seasonal cycle, and anomalies from this in the imputation methods. | 2 | This point is also well taken. With removing the CO2 flux aspect of the paper we can provide more room for showing the distribution of error. As for the structure of the seasonal cycle, we disucss this but had not quantified it. In revision we will provide quanitification of the timing and magnitude of the seasonal cycle and some metric(s) for method performance to make this discussion less qualitative. |
| 26 | NA | The use of the air-sea CO2 flux for assessing imputation performance is an interesting choice, as it introduces a whole suite of additional uncertainties (wind-speed, piston velocity, K1/K2 equilibrium constants, how missing alkalinity data is filled etc) that are not considered in your error analysis. These uncertainties would also need to be assessed, or another metric/s chosen for comparison. If the air-sea CO2 flux is still the preferred metric, is it not better to calculate pCO2 from DIC/alkalinity first, before imputing missing pCO2 values? | 2 | See our response to reviewer 1 comments above regarding our initial methods and reasoning; and note that we will be removing this aspect from the paper. |
| 27 | NA | I appreciate that this may be being considered in a follow up study, but an assessment of the desired sampling frequency necessary to generate a good representation of the seasonal cycle (1, 1.5, 2, 3 month frequency, only summer and winter etc) would be very interesting/useful. | 2 | We will add this assessment |
| 28 | 36 | value is singular, so has not have | 2 | This has been addressed |
| 29 | 38 | 40% - This is possibly fossil fuel CO2 emissions? All anthropogenic CO2 (including land-use change and cement) means the ocean component is probably closer to 25% (Global Carbon Project, Friedlingstein et al., 2020) | 2 | This has been addressed |

| | 66 | "This study aims to identify the optimal gap-filling methods for carbonate time series by establishing which techniques perform with sufficiently low error and bias to assess seasonal and interannual variability of carbonate biogeochemistry and the biological and physical processes that determine it." - see comment above | 2 | Response as above |
|---|---|---|---|---|
| 30 | | | | |
| 31 | 72 | should be principle rather than principal | 2 | This has been addressed |
| 32 | 75 | (and Table 1) - add citation/references for time-series, possibly through additional column in Table | 2 | As per response to Adrinne's comment above, this was an oversight and all dataset citations will be properly added. |
| 33 | 86 | Is there an impact on your analyses of averaging data to monthly means? | 2 | Uncertainty in monthly values was estimated for both single observations and averaged higher frequency measurements from moorings so they could be properly compared. We will make sure this is clearly communicated in the methdos during revision |
| 34 | 89 | would be better to use greek delta notation rather than DT | 2 | fixed per above as well |
| 35 | 90 | What is the uncertainty introduced by the use of estimated DIC values? DIC is only measured at BATS. What do you get if you apply the same techniques to data with DIC, TA and pCO2 e.g. at sea surface? | 2 | Individual DIC uncertainty budgets were assess by adding the sources (measurement, natural variability (e.g. monthly averaging), and /or propogation from calculating DIC from other carbonate measurements) in quadrature to determine the combined statndard uncertainty for each DIC value in the time series. For DIC calculated from the other variabiles such as pCO2 and TA, the error function in the R package seacarb was used. |
| 36 | 122 | "The primary goal was imputing timeseries at monthly resolution to investigate variability and trends over seasonal, interannual and decadal timescales" - neither trends nor decadal are covered as far as I can see? | 2 | See our responses above that indicate we will be removing the CO2 flux aspect and adding an assessment of trends and seasonal structure |
| 37 | 141 | is this not an exponential moving average then, rather than a weighted moving average? | 2 | I suppose it could be stated both ways. It is a weighted moving average, but the weighting is based on an exponential relation to neighbors |
| 38 | 148 | cite1 and cite2? | 2 | This was some sort of formating typo with Endnote - will fix |
| 39 | 150 | does this method also input uncertainty into the fitted values used? | 2 | I don't believe this inputs uncertainty - rather values are found through convergence of multiple regressions. Unertainty can be assessed by looking at the spread when the option to have multple outputs for a give value is selected. |
| 40 | 190 | as above, why this? Is it not better to calculate pCO2 from bottles at the start, then do imputation on pCO2 data set? | 2 | No imputation of pCO2 data was done. All imputation is on DIC values only. All pCO2 was calucated from the imputed DIC and either measured or estiamted alkalinity |
| 41 | 193 | Wanninkhof 2014 recommends to not use Wanninkhof 1992. | 2 | removing CO2 flux aspect |
| 42 | 201 | why not use Bermuda atmospheric CO2 concentrations? | 2 | removing CO2 flux aspect |
| 43 | 215 | what were these uncertainties? It would be good to state them here. pCO2 from DIC and TA at their measurement uncertainty is ~6uatm. What is it when DIC is estimated? | 2 | We will make unceraities more explicit during revision |
| 44 | 223 | To give a better feeling of interannual variability it would be useful to have the value for n for each month in Figure 2. For example so that a reader doesn't look at FOT and think there is very little variability in months 1-3, when instead n is only 1-2 for these months. | 2 | will add this info |
| 45 | 227 | & Fig 3. Is this a single MLR encompassing all data from all sites? Or the results of individual MLRs plotted and pooled? I'm don't think this is clear in the text | 2 | This is pooled results. MLRs must be fit using site specific observations and have unique coefficients. Will update language to clarify |
| 46 | 229 | "worked well"? A RMSE of 12 is beyond the 'weather' goal of measurement quality to assess spatial and short-term variability. I'm not sure stating this metric is useful as it obscures the capability of the method in (primarily) oceanic sites. Instead it might be better to simply focus on individual monitoring station results. | 2 | This is a good point and we will address during revision |

| | 234 | It would be interesting to hear the thoughts behind why PAPA performs so well | 2 | The MLR appears to perform best at sites where there is a high correlation to temperature and a large seasonal cycle. The performance of the results follows the trends shown in Figure 5 where there is selective ommission of predictor variables. We will investigate and elaborate on this further during revision |
|---|---|---|---|---|
| 47 | | | | |
| 48 | 244 | put the numbers in the boxes as well - the colour scale is not the most obvious/immediate to show similarity/disparity | 2 | We will add this info |
| 49 | 245 | add another line to the bottom of Figure 5 to show mean | 2 | We will add this info |
| 50 | 246 | Table 5 - change title to Mean model results | 2 | This has been addressed |
| 51 | 250 | Figure 6 - might be better showing as well / instead the residual (y) versus the measured (x)? - this may better highlight the better performing models, with the distribution of the residual ideally normal about 0. | 2 | We will explore this suggestion and other possiblities for expressing error distribution across both observed DIC ranges and sites |
| 52 | 259 | I struggle somewhat with this plot (Fig 7) too. The colour scale is not the most obvious/immediate to show similarity/disparity, and seems to be the opposite to Figure 5 where light colours indicate better performance - here they indicate worse performance. | 2 | We will look to increase the contrast and make these figure gradients consistent for clarity |
| 53 | 261 | I think that showing the performance of the models in recreating the seasonal cycle would be very useful. Whether they get the amplitude and timing correct is important for potential end users of these methods. Showing the anomaly from the observed seasonal cycle may also be useful. | 2 | This is a great point. As indicated in response above, this was qualified in the discussion but was lacking quantification and we will add that during revision |
| 54 | 266 | Fig 8A I like this plot, but i think it is making false equivalences by using different y scales for the 7 different methods for each monitoring station. It might be worth having this as a standalone figure to give more space to what is an enormous amount of information. | 2 | With the removal of the CO2 flux aspect and associate figure we will have space to break out this figure. The y scales were held consistent across sites so that methods could be compared. If the scales are held constant for all sites and all methods it will loose significant detail for visual interpretation. |
| 55 | 275 | Assessing error on seasonality and annual sums - not sure these numbers capture this. As mentioned above I'd be interested in seeing the performance of individual methods of capturing the seasonal cycle / amplitude and annual mean, and how they compare to the data, both using the full timeseries, and when there are artifical data gaps. It would certainly be useful to know how critical it is to sample seasonal maxima/minima (or not) in correctly formulating a seasonal cycle, and getting lowering the uncertainty with respect to annual budgets. | 2 | The aspect of sampling optimization is a good point that is missing here. As noted in responses above, we will quantify the performance of retaining seasonal structure and we will explore assessing sampling optimization. |
| 56 | 280 | and Figure 9A. While these plots are interesting it might be better represented by adding/replacing wih anomaly timeseries. Also, I was wondering whether you could comment on how there appears to be a positive bias for the bimonthly and 3 month data gaps towards higher concentrations? Is the reason there are no red dots at the lowest concentrations (particularly in the 3 month timescale) simply the result of random data gaps, or something else? For the 6 month gaps I'd be interested in the performance of the models when only summer data is available, or perhaps completely missing winter data, as this would be a situaion facing other time series sites. | 2 | We can explore representing anomalies here for clarity. As for the gap placement in the 3-month gap series, yes this is just due to randomization. We could explore artificially removing particular seasons and assessing impacts on annual cycles. |
| 57 | 291 | Fig 9b - would it be possible to have the legend across a single row, to aid in identifying models? Or indeed numbering the different box plots. | 2 | We will address clarifying the identification of methods in this boxplot |
| 58 | 299 | Figure 10 - this plot mght be easier to interpret if it was anomalies from observations rather than actual values side-by-side? | 2 | This figure will be removed along with the CO2 flux aspect of the paper |

| | | | | |
|---|---|---|---|---|
| 59 | | The uncertainty bars also seem particularly low - has the uncertainty from the imputed data been propagated through the calculation? Even a DIC RMSE of 6 umol/kg would have an impact of 10-25uatm of pCO2 depending on temperature. I imagine if there are missing DIC observations, there will also be missing alkalinity observations as well. It will likely be too much to include an estimate from these values as well, but I think you should comment on the fact that the error estimates relating to air-sea CO2 fluxes presented here will be an underestimate, as there will also be addiional uncertainties associated with imputing alkalinity. | 2 | The uncertainty budget was assessed using a MCM method as noted above in response to other comments, however we will be removing this aspect regardless in place of more focus on assessing trends and seasonal strcuture |
| 60 | 328 | change 'has a dominant effect the carbonate chemistry' to 'has a dominant effect on carbonate chemistry' | 2 | This has been addressed |
| 61 | 333 | need to referencce these different datasets | 2 | This will be address as noted above |
| 62 | 335 | missing full stop | 2 | This has been addressed |
| 63 | 353 | - I don't think you've shown anything about temporal extrapolation. | 2 | This has been addressed |
| 64 | 358 | either remove the parentheses around the citations, or remove 'in the studies of' | 2 | This has been addressed - this was an Endnote formatting typo |
| 65 | 369 | This may be so but I don't think the figures you have presented make this obvious. A figure showing the mean seasonal cyle from the full data set compared to those imputed for different percentages of missing data would be necessary to show this. | 2 | Our quantification of seasonal structure during revision will address this |
| 66 | 371 | it's not clear visually, as you're missing a figure showing it. Figure 9 suggests it's only really obvious for the 6 month gap, while Figure 12 suggests that the mean approach has some of the highest uncertainties for the bi-monthly data gaps. | 2 | Our quantification of seasonal structure during revision will address this |
| 67 | 381 | I'd again suggest that looking at anomaly plots would be more straightforward to interpret than net flux comparisons | 2 | Point well taken, and we will explore this for clarity |
| 68 | 405 | change 'In general' to 'Of the methods we tested' | 2 | This has been addressed |
| 69 | 408 | May and possibly are really not strong enough - the artifice of the mean imputation method introduces bias, and actively removes any trend from the input data. | 2 | Good point, we will revise language here |
| 70 | 415 | - MLR certainly has the lowest error, but this doesn't necessarily tell the whole story. Showing the residuals of the predicted values will help - would you like to comment on the tendency of MLR methods to revert to the mean, where higher values are typically predicted lower, and lower values are predicted higher. This will have an impact on estimating maxima/minima. And I'd hesitate to recommend best practice until MLR is compared against a fuller suite of gap-filling methods, including machine learning | 2 | As also noted above in responses to a similar comment, we will revise the focus of the paper to dial back the language for establishing best practices and shift to scoping it as a presentation of this MLR compared to some selected common methods. The expanded seasonal structure assessment will help the discussion about max/min biasing |
| 71 | 426 | (and L433)- can be estimated, but to what uncertainty, and is this the same across all times of the year? | 2 | We will assess and present seasonal error distribution to futher support this; however uncertainty must be assess on an individual site/data set basis. We will make revisions to the language here to make sure that point is clear |
| 72 | 432 | I sound like a broken record but I think plots of seaonal cycles/anoamlies of seasonal cycles/internannual anaomlies are really what are needed to help determine this. | 2 | Noted and will address |
| 73 | 433 | Change "the most robust option for imputing gaps over a variety of data gap scenarios." to "the most robust option from those we compared for imputing gaps over a variety of data gap scenarios." | | This has been addressed |

---

## Author Response (AR1)

| Line (initial) | Comment | Reviewer | Response (revision or comment) | Revised Line |
|---|---|---|---|---|
| Section 2.6 | For the error propagation described in section 2.6, what is the reasoning for using CO2 flux? Using CO2 flux introduces several other potential biases and errors to the assessment: | 1 | The initial reasoning for inlcuding the CO2 flux was to show implications of error propogation for various imputations during a common use case for DIC time series. However, both reviewers have raised similar concern about the introduction of multiple sources of error when determining CO2 flux. The combined uncertainty for the CO2 flux was initially determined by a Monte Carlo method (n=1000, which was not significantly different than n=10000) and then only the values of imputed DIC and their associeted uncertainty were varied as inputs into the calculation. In this way we attributed the percent difference between imputed time series and observed time series to be related only to the gap-filling method because no other input was varied. Similarly the uncertainty of the CO2 flux was determined via this MCM for each method. That said, we understand the concern about multiple sources of error and recognize that this application detracts from the results of just gap-filling the DIC time series. We will remove the CO2 flux aspect of paper and add a focus on long term trend assessment in its place. This will be more consistent with the intentions of the work, enhance the focus of the paper and address mulitple comments from both reviewers. | Section 2.7 inluces the updated uncertainty budget |
| | · uncertainty in air pCO2, | 1 | removing CO2 flux aspect | |
| | · major bias and errors of NCEP winds (see: https://doi.org/10.5194/bg-15-1701-2018, https://doi.org/10.1029/2018GB006047, https://doi.org/10.1002/2017GL073814) | 1 | removing CO2 flux aspect | NA |
| | · uncertainty in the gas transfer velocity coefficient (resulting in total uncertainty in CO2 flux of ~20%), and | 1 | removing CO2 flux aspect | NA |
| | · uncertainty (~5%) introduced in the calculation of sw pCO2 from DIC and TA. | 1 | removing CO2 flux aspect | NA |
| | How will those biases and errors complicate your assessment of gap filling error propagation? The relative uncertainty for CO2 flux at BATS is reported in line 361 as 3.5%. What does this uncertainty take into account? Not items 1 – 4 above, as this value would be much higher. These issues should be addressed in the error propagation, or another parameter should be used for this assessment. | 1 | removing CO2 flux aspect | NA |
| NA | Data used in this study need to be cited properly, which is incredibly important to the programs supporting these time series measurements. Those data should be cited in the methods and/or funders noted in the acknowledgements, depending on what each time series program recommends, not recorded as web addresses in the notes of Table 2. For the moorings, if you are accessing original data files via NCEI, those citations can be found at https://doi.org/10.3334/cdiac/otg.tsm_papa_145w_50n for Papa and https://doi.org/10.3334/cdiac/otg.tsm_keo_145e_32n for KEO. If you are accessing the mooring data from the synthesis product, the citation can be found at https://doi.org/10.7289/V5DB8043. I am not as familiar with the citation requirements of all the ship-based time series, but with a quick search I found this data citation request for HOTS, for example: https://hahana.soest.hawaii.edu/hot/dataaccess.html | 1 | This was a gross oversight on our part. While the sources for data sets were listed in table 1, they were not properly cited as noted. We will cite these as required. | See Section 2.1 Field Data lines 85-95 and Acknowledgements |
| NA | Finally, it may be out of the scope to include additional analyses in this paper, but it would be worthwhile discussing future work that can build off these results. For example, what satellite-based products are best suited for the MLR approach? Are there any that can span open ocean and coastal environments, so gap filling methods can be applied consistently across all global ocean and coastal time series? Also, it would be useful to study whether there are discrepancies in calculated trends when using these different gap filling methods (at least the most successful methods) or no gap filling methods at all. Both of these analyses seem like they could have been included in this paper, but I could also understand if those are the next assessments planned using the most promising empirical gap filling methods resulting from this work. | 1 | These are excellent points some of which we can address in the revision. Firstly, we will include an assessment of impacts on trends in place of the CO2 flux. Secondly, we have already separately performed a cross shelf assessment of the MLR performance using data from the Munida transect and we can included this appliaction. These aspect taken together will also help address Reviewer 2's comment about focusing the scope of the paper more on presenting this MLR method and comparing it to other gap-filling methods, rather than an extensive comparitive assessment of techniques since we have only selected a few methods from a very large number of possiblities. | spatial extrapolation is not included in the current scope but future developoments are discussed through section 4.2 and 5 |
| Line (initial) | Comment | Reviewer | Response (revision or comment) | Revised Line |
| Section 2.7 | For the error propagation described in section 2.6, what is the reasoning for using CO2 flux? Using CO2 flux introduces several other potential biases and errors to the assessment: | 1 | The initial reasoning for inlcuding the CO2 flux was to show implications of error propogation for various imputations during a common use case for DIC time series. However, both reviewers have raised similar concern about the introduction of multiple sources of error when determining CO2 flux. The combined uncertainty for the CO2 flux was initially determined by a Monte Carlo method (n=1000, which was not significantly different than n=10000) and then only the values of imputed DIC and their associeted uncertainty were varied as inputs into the calculation. In this way we attributed the percent difference between imputed time series and observed time series to be related only to the gap-filling method because no other input was varied. Similarly the uncertainty of the CO2 flux was determined via this MCM for each method. That said, we understand the concern about multiple sources of error and recognize that this application detracts from the results of just gap-filling the DIC time series. We will remove the CO2 flux aspect of paper and add a focus on long term trend assessment in its place. This will be more consistent with the intentions of the work, enhance the focus of the paper and address mulitple comments from both reviewers. | Section 2.7 inluces the updated uncertainty budget |
| | · uncertainty in air pCO2, | 1 | removing CO2 flux aspect | |
| | · major bias and errors of NCEP winds (see: https://doi.org/10.5194/bg-15-1701-2018, https://doi.org/10.1029/2018GB006047, https://doi.org/10.1002/2017GL073814) | 1 | removing CO2 flux aspect | NA |
| | · uncertainty in the gas transfer velocity coefficient (resulting in total uncertainty in CO2 flux of ~20%), and | 1 | removing CO2 flux aspect | NA |
| | · uncertainty (~5%) introduced in the calculation of sw pCO2 from DIC and TA. | 1 | removing CO2 flux aspect | NA |
| | How will those biases and errors complicate your assessment of gap filling error propagation? The relative uncertainty for CO2 flux at BATS is reported in line 361 as 3.5%. What does this uncertainty take into account? Not items 1 – 4 above, as this value would be much higher. These issues should be addressed in the error propagation, or another parameter should be used for this assessment. | 1 | removing CO2 flux aspect | NA |

| Line (initial) | Comment | Reviewer | Response (revision or comment) | Revised Line |
|---|---|---|---|---|
| NA | Data used in this study need to be cited properly, which is incredibly important to the programs supporting these time series measurements. Those data should be cited in the methods and/or funders noted in the acknowledgements, depending on what each time series program recommends, not recorded as web addresses in the notes of Table 2. For the moorings, if you are accessing original data files via NCEI, those citations can be found at https://doi.org/10.3334/cdiac/otg.tsm_papa_145w_50n for Papa and https://doi.org/10.3334/cdiac/otg.tsm_keo_145e_32n for KEO. If you are accessing the mooring data from the synthesis product, the citation can be found at https://doi.org/10.7289/V5DB8043. I am not as familiar with the citation requirements of all the ship-based time series, but with a quick search I found this data citation request for HOTS, for example: https://hahana.soest.hawaii.edu/hot/dataaccess.html | 1 | This was a gross oversight on our part. While the sources for data sets were listed in table 1, they were not properly cited as noted. We will cite these as required. | See Section 2.1 Field Data lines 85-95 and Acknowledgements |
| NA | Finally, it may be out of the scope to include additional analyses in this paper, but it would be worthwhile discussing future work that can build off these results. For example, what satellite-based products are best suited for the MLR approach? Are there any that can span open ocean and coastal environments, so gap filling methods can be applied consistently across all global ocean and coastal time series? Also, it would be useful to study whether there are discrepancies in calculated trends when using these different gap filling methods (at least the most successful methods) or no gap filling methods at all. Both of these analyses seem like they could have been included in this paper, but I could also understand if those are the next assessments planned using the most promising empirical gap filling methods resulting from this work. | 1 | These are excellent points some of which we can address in the revision. Firstly, we will include an assessment of impacts on trends in place of the CO2 flux. Secondly, we have already separately performed a cross shelf assessment of the MLR performance using data from the Munida transect and we can included this appliaction. These aspect taken together will also help address Reviewer 2's comment about focusing the scope of the paper more on presenting this MLR method and comparing it to other gap-filling methods, rather than an extensive comparitive assessment of techniques since we have only selected a few methods from a very large number of possiblities. | spatial extrapolation is not included in the current scope but future developoments are discussed through section 4.2 and 6 |
| Line (initial) | Comment | Reviewer | Response (revision or comment) | Revised Line |
| Section 2.8 | For the error propagation described in section 2.6, what is the reasoning for using CO2 flux? Using CO2 flux introduces several other potential biases and errors to the assessment: | 1 | The initial reasoning for inlcuding the CO2 flux was to show implications of error propagation for various imputations during a common use case for DIC time series. However, both reviewers have raised similar concern about the introduction of multiple sources of error when determining CO2 flux. The combined uncertainty for the CO2 flux was initially determined by a Monte Carlo method (n=1000, which was not significantly different than n=10000) and then only the values of imputed DIC and their associeted uncertainty were varied as inputs into the calculation. In this way we attributed the percent difference between imputed time series and observed time series to be related only to the gap-filling method because no other input was varied. Similarly the uncertainty of the CO2 flux was determined via this MCM for each method. That said, we understand the concern about multiple sources of error and recognize that this application detracts from the results of just gap-filling the DIC time series. We will remove the CO2 flux aspect of paper and add a focus on long term trend assessment in its place. This will be more consistent with the intentions of the work, enhance the focus of the paper and address mulitple comments from both reviewers. | Section 2.7 inlcues the updated uncertainty budget |
| | · uncertainty in air pCO2, | 1 | removing CO2 flux aspect | |
| | · major bias and errors of NCEP winds (see: https://doi.org/10.5194/bg-15-1701-2018, https://doi.org/10.1029/2018GB006047, https://doi.org/10.1002/2017GL073814) | 1 | removing CO2 flux aspect | NA |
| | · uncertainty in the gas transfer velocity coefficient (resulting in total uncertainty in CO2 flux of ~20%), and | 1 | removing CO2 flux aspect | NA |
| | · uncertainty (~5%) introduced in the calculation of sw pCO2 from DIC and TA. | 1 | removing CO2 flux aspect | NA |
| 66 | On Line 66 is stated "This study aims to identify the optimal gap-filling methods for carbonate time series by establishing which techniques perform with sufficiently low error and bias to assess seasonal and interannual variability of carbonate biogeochemistry and the biological and physical processes that determine it." The manuscript takes the approach that all gap-filling techniques have been explored and that MLR is recommended as the best performing. While the latter is certainly true of the methods compared, I feel it is not currently possible to say the former while one / a number of machine learning (and other) approaches are absent - these have recently been successfully applied in oceanographic research, and so the manuscript is not fulfilling its own aims by omitting them. Clearly it is not feasible to compare all available methodologies, so I would recommend that you either tone down the aims of the paper (by saying that you present a MLR method for DIC time-series data gap imputation and compare it to other common, computationally inexpensive methods) or a selection of additional methods are included e.g. median as well as mean, machine learning (i.e. neural network, regression trees, random forests that you already mention), curve fitting, exponential moving average, k-nearest neighbours etc. | 2 | This point is well-taken. Given that we have not here (and could not really) assessed all methods, we will shift the stated focus away from optimization of gap-filling and toward presenting the MLR and comparing it against other common approaches as suggested. | Lines 77-82 |
| NA | When comparing methods a lot of focus is on the magnitude of the RMSE. I feel the reader would benefit from some consideration of the structure of the error e.g. are certain times of the year subject to greater uncertainties, do the models reproduce the timing of the seasonal cycle, and the magnitude of the peaks and troughs or are these far worse than those that vary around annual mean values? Equally, is the error of the preferred MLR technique actually normally distributed, as a lot of its power rests on this assumption. The manuscript would certainly benefit from greater examination of the seasonal cycle, and anomalies from this in the imputation methods. | 2 | This point is also well taken. With removing the CO2 flux aspect of the paper we can provide more room for showing the distribution of error. As for the structure of the seasonal cycle, we disucss this but had not quantified it. In revision we will provide quanitification of the timing and magnitude of the seasonal cycle and some metric(s) for method performance to make this discussion less qualitative. | This is address through the rivsed results and discussion sections |
| NA | The use of the air-sea CO2 flux for assessing imputation performance is an interesting choice, as it introduces a whole suite of additional uncertainties (wind-speed, piston velocity, K1/K2 equilibrium constants, how missing alkalinity data is filled etc) that are not considered in your error analysis. These uncertainties would also need to be assessed, or another metric/s chosen for comparison. If the air-sea CO2 flux is still the preferred metric, is it not better to calculate pCO2 from DIC/alkalinity first, before imputing missing pCO2 values? | 2 | See our response to reviewer 1 comments above regarding our initial methods and reasoning; and note that we will be removing this aspect from the paper. | NA |

| | | | | |
|---|---|---|---|---|
| NA | I appreciate that this may be being considered in a follow up study, but an assessment of the desired sampling frequency necessary to generate a good representation of the seasonal cycle (1, 1.5, 2, 3 month frequency, only summer and winter etc) would be very interesting/useful. | 2 | We will add this assessment | Updated by inclusion of various gap filters and address through revised results and discussion secitons |
| 36 | value is singular, so has not have | 2 | This has been addressed | Line 46 |
| 38 | 40% - This is possibly fossil fuel CO2 emissions? All anthropogenic CO2 (including land-use change and cement) means the ocean component is probably closer to 25% (Global Carbon Project, Friedlingstein et al., 2020) | 2 | This has been addressed | Line 49 |
| 66 | "This study aims to identify the optimal gap-filling methods for carbonate time series by establishing which techniques perform with sufficiently low error and bias to assess seasonal and interannual variability of carbonate biogeochemistry and the biological and physical processes that determine it." - see comment above | 2 | Response as above | Lines 77-82 |
| 72 | should be principle rather than principal | 2 | This has been addressed | Line 82 |
| 75 | (and Table 1) - add citation/references for time-series, possibly through additional column in Table | 2 | As per response to Adrinne's comment above, this was an oversight and all dataset citations will be properly added. | Addresed in section 2.1 and Acknowledgements |
| 86 | Is there an impact on your analyses of averaging data to monthly means? | 2 | Uncertainty in monthly values was estimated for both single observations and averaged higher frequency measurements from moorings so they could be properly compared. We will make sure this is clearly communicated in the methdos during revision | Included in uncertainty budget now - see section 2.7 |
| 89 | would be better to use greek delta notation rather than DT | 2 | fixed per above as well | line 99 |
| 90 | What is the uncertainty introduced by the use of estimated DIC values? DIC is only measured at BATS. What do you get if you apply the same techniques to data with DIC, TA and pCO2 e.g. at sea surface? | 2 | Individual DIC uncertainty budgets were assess by adding the sources (measurement, natural variability (e.g. monthly averaging), and /or propogation from calculating DIC from other carbonate measurements) in quadrature to determine the combined statndard uncertainty for each DIC value in the time series. For DIC calculated from the other variabiles such as pCO2 and TA, the error function in the R package seacarb was used. | See Section 2.7 for updated uncertainty |
| 122 | "The primary goal was imputing timeseries at monthly resolution to investigate variability and trends over seasonal, interannual and decadal timescales" - neither trends nor decadal are covered as far as I can see? | 2 | See our responses above that indicate we will be removing the CO2 flux aspect and adding an assessment of trends and seasonal structure | trends and other seasonal and interannual variability were quantified in the results and discussed in section 4.2 |
| 141 | is this not an exponential moving average then, rather than a weighted moving average? | 2 | I suppose it could be stated both ways. It is a weighted moving average, but the weighting is based on an exponential relation to neighbors | It is more cldearly referred to as exponential wma |
| 148 | cite1 and cite2? | 2 | This is some sort of formating typo with Endnote - will fix | Line 173 |
| 150 | does this method also input uncertainty into the fitted values used? | 2 | I don't believe this inputs uncertainty - rather values are found through convergence of multiple regressions. Unertainty can be assessed by looking at the spread when the option to have multple outputs for a give value is selected. | NA |
| 190 | as above, why this? Is it not better to calculate pCO2 from bottles at the start, then do imputation on pCO2 data set? | 2 | No imputation of pCO2 data was done. All imputation is on DIC values only. All pCO2 was caluacted from the imputed DIC and either measured or estiamted alkalinity | NA |
| 193 | Wanninkhof 2014 recommends to not use Wanninkhof 1992. | 2 | removing CO2 flux aspect | NA |
| 201 | why not use Bermuda atmospheric CO2 concentrations? | 2 | removing CO2 flux aspect | NA |
| 215 | what were these uncertainties? It would be good to state them here. pCO2 from DIC and TA at their measurement uncertainty is ~6uatm. What is it when DIC is estimated? | 2 | We will make unceraintes more explicity during revision | See Section 2.7 for updated uncertainty |
| 223 | To give a better feeling of interannual variability it would be useful to have the value for n for each month in Figure 2. For example so that a reader doesn't look at FOT and think there is very little variability in months 1-3, when instead n is only 1-2 for these months. | 2 | will add this info | Updated Fig. 2 |
| 227 | & Fig 3. Is this a single MLR encompassing all data from all sites? Or the results of individual MLRs plotted and pooled? I'm don't think this is clear in the text | 2 | This is pooled results. MLRs must be fit using site specific observations and have unique coefficients. Will update language to clarify | Updated to Fig caption (Now Fig 4) |
| 229 | "worked well"? A RMSE of 12 is beyond the 'weather' goal of measurement quality to assess spatial and short-term variability. I'm not sure stating this metric is useful as it obscures the capability of the method in (primarily) oceanic sites. Instead it might be better to simply focus on individual monitoring station results. | 2 | This is a good point and we will address during revision | updated these points in results and in discussion section 4.2 |
| 234 | It would be interesting to hear the thoughts behind why PAPA performs so well | 2 | The MLR appears to perform best at sites where there is a high correlation to temperature and a large seasonal cycle. The performance of the results follows the trends shown in Figure 5 where there is selective ommission of predictor variables. We will investigate and elaborate on this further during revision | Line 290 |
| 244 | put the numbers in the boxes as well - the colour scale is not the most obvious/immediate to show similarity/disparity | 2 | We will add this info | updated tile figures now Figs 6 & 8 |
| 245 | add another line to the bottom of Figure 5 to show mean | 2 | We will add this info | updated figure 6 |
| 246 | Table 5 - change title to Mean model results | 2 | This has been addressed | Line 932 |
| 250 | Figure 6 - might be better showing as well / instead the residual (y) versus the measured (x)? - this may better highlight the better performing models, with the distribution of the residual ideally normal about 0. | 2 | We will explore this suggestion and other possiblities for expressing error distribution across both observed DIC ranges and sites | Updated Figure to show residuals as kernel density plots as this seems to provide the best representation for the point that was being asked here. Basic residual plots were visually messy and did not provide additional clarity. |
| 259 | I struggle somewhat with this plot (Fig 7) too. The colour scale is not the most obvious/immediate to show similarity/disparity, and seems to be the opposite to Figure 5 where light colours indicate better performance - here they indicate worse performance. | 2 | We will look to increase the contrast and make these figure gradients consistent for clarity | updated tile figures for consistency |
| 261 | I think that showing the performance of the models in recreating the seasonal cycle would be very useful. Whether they get the amplitude and timing correct is important for potential end users of these methods. Showing the anomaly from the observed seasonal cycle may also be useful. | 2 | This is a great point. As indicated in response above, this was qualified in the discussion but was lacking quantification and we will add that during revision | Created new analyses and metrics to quantify this and included in results and discussion sections |
| 266 | Fig 8A I like this plot, but i think it is making false equivalences by using different y scales for the 7 different methods for each monitoring station. It might be worth having this as a standalone figure to give more space to what is an enormous amount of information. | 2 | With the removal of the CO2 flux aspect and associate figure we will have space to break out this figure. The y scales were held consistent across sites so that methods could be compared. If the scales are held constant for all sites and all methods it will loose significant detail for visual interpretation. | The y scales are consistent for sites. This figure was also updated to make it slightly cleaner for visibility |
| 275 | Assessing error on seasonality and annual sums - not sure these numbers capture this. As mentioned above I'd be interested in seeing the performance of individual methods of capturing the seasonal cycle / amplitude and annual mean, and how they compare to the data, both using the full timeseries and when there are artifical data gaps. It would certainly be useful to know how critical it is to sample seasonal maxima/minima (or not) in correctly formulating a seasonal cycle, and getting lowering the uncertainty with respect to annual budgets. | 2 | The aspect of sampling optimization is a good point that is missing here. As noted in responses above, we will quantify the performance of retaining seasonal structure and we will explore assessing sampling optimization. | Created new analyses and metrics to quantify this and included in results and discussion sections |

| | Comment | | Response | Location |
|---|---|---|---|---|
| 280 | and Figure 9A. While these plots are interesting it might be better represented by adding/replacing wih anomaly timeseries. Also, I was wondering whether you could comment on how there appears to be a positive bias for the bimonthly and 3 month data gaps towards higher concentrations? Is the reason there are no red dots at the lowest concentrations (particularly in the 3 month timescale) simply the result of random data gaps, or something else? For the 6 month gaps I'd be interested in the performance of the models when only summer data is available, or perhaps completely missing winter data, as this would be a situtaion facing other time series sites. | 2 | We can explore representing anomalies here for clarity. As for the gap placement in the 3-month gap series, yes this is just due to randomization. We could explore artificially removing particular seasons and assessing impacts on annual cycles. | NA |
| 291 | Fig 9b - would it be possible to have the legend across a single row, to aid in identifying models? Or indeed numbering the different box plots. | 2 | We will address clarifying the identification of methods in this boxplot | NA |
| 299 | Figure 10 - this plot mght be easier to interpret if it was anomalies from observations rather than actual values side-by-side? | 2 | This figure will be removed along with the CO2 flux aspect of the paper | NA |
| | The uncertainty bars also seem particularly low - has the uncertainty from the imputed data been propagated through the calculation? Even a DIC RMSE of 6 umol/kg would have an impact of 10-25uatm of pCO2 depending on temperature. I imagine if there are missing DIC observations, there will also be missing alkalinity observations as well. It will likely be too much to include an estimate from these values as well, but I think you should comment on the fact that the error estimates relating to air-sea CO2 fluxes presented here will be an underestimate, as there will also be addiional uncertainties associated with imputing alkalinity. | 2 | The uncertainty budget was assessed using a MCM method as noted above in response to other comments, however we will be removing this aspect regardless in place of more focus on assessing trends and seasonal strcuture | NA |
| 328 | change 'has a dominant effect the carbonate chemistry' to 'has a dominant effect on carbonate chemistry' | 2 | This has been addressed | Line 456 |
| 333 | need to referencce these different datasets | 2 | This will be address as noted above | See Section 2.1 Field Data lines 85-95 and Acknowledgements |
| 335 | missing full stop | 2 | This has been addressed | Line 464 |
| 353 | - I don't think you've shown anything about temporal extrapolation. | 2 | This has been addressed | Line 484 |
| 358 | either remove the parentheses around the citations, or remove 'in the studies of' | 2 | This has been addressed - this was an Endnote formatting typo | Line 498 |
| 369 | This may be so but I don't think the figures you have presented make this obvious. A figure showing the mean seasonal cyle from the full data set compared to those imputed for different percentages of missing data would be necessary to show this. | 2 | Our quantification of seasonal structure during revision will address this | See Fig 14 with results and discussion revisions |
| 371 | it's not clear visually, as you're missing a figure showing it. Figure 9 suggests it's only really obvious for the 6 month gap, while Figure 12 suggests that the mean approach has some of the highest uncertainties for the bi-monthly data gaps. | 2 | Our quantification of seasonal structure during revision will address this | See Fig 14 with results and discussion revisions |
| 381 | I'd again suggest that looking at anomaly plots would be more straightforward to interpret than net flux comparisons | 2 | Point well taken, and we will explore this for clarity | NA |
| 405 | change 'In general' to 'Of the methods we tested' | 2 | This has been addressed | Line 573 |
| 408 | May and possibly are really not strong enough - the artifice of the mean imputation method introduces bias, and actively removes any trend from the input data. | 2 | Good point, we will revise language here | Line 575 |
| 415 | - MLR certainly has the lowest error, but this doesn't necessarily tell the whole story. Showing the residuals of the predicted values will help - would you like to comment on the tendency of MLR methods to revert to the mean, where higher values are typically predicted lower, and lower values are predicted higher. This will have an impact on estimating maxima/minima. And I'd hesitate to recommend best practice until MLR is compared against a fuller suite of gap-filling methods, including machine learning | 2 | As also noted above in responses to a similar comment, we will revise the focus of the paper to dial back the language for establishing best practices and shift to scoping it as a presentation of this MLR compared to some selected common methods. The expanded seasonal structure assessment will help the discussion about max/min biasing | Lines 585-591 |
| 426 | (and L433)- can be estimated, but to what uncertainty, and is this the same across all times of the year? | 2 | We will assess and present seasonal error distribution to futher support this; however uncertainty must be assess on an individual site/data set basis. We will make revisions to the language here to make sure that point is clear | Lines 604 - 612 |
| 432 | I sound like a broken record but I think plots of seaonal cycles/anoamlies of seasonal cycles/internannual anaomlies are really what are needed to help determine this. | 2 | Noted and will address | See Fig 14 with results and discussion revisions |
| 433 | Change "the most robust option for imputing gaps over a variety of data gap scenarios." to "the most robust option from those we compared for imputing gaps over a variety of data gap scenarios." | | This has been addressed | Lines 604 - 612 |

---

## Author Response (AR2)

| Line (initial) | Comment | Reviewer | Response (revision or comment) | Revised Line |
|---|---|---|---|---|
| Section 2.6 | In section 2.6 it is unclear how the trends are calculated. A linear regression of the imputed and observed time series of monthly means? If so, the results may not be applicable to the most common approach currently used in the ocean carbon community, which is to apply a linear regression to deseasoned monthly means, where seasonal variability is removed (Bates 2001; Bates et al. 2014; Takahashi et al. 2009). In this approach, gaps are filled if months are missing in the climatological monthly means used to remove the seasonal signal (see Figure 2 in Takahashi et al. 2009), but data gaps within the resulting time series of deseasoned monthly means are not (see Figure 3 in Takahashi et al. 2009). If the trends presented by Vance et al. are on data that include additional noise from seasonal variability, would the impact of different gap-filling techniques on resulting trends be the same for the deseasoned approach where seasonal variability is removed? | 1 | We have applied the method of Takahashi et al. 2009 to seasonally detrend the time series before applying the linear regression to establish the long term ternds for observations and comparison to imputed time series. With this we have updated Section 2.6, adding Equation 10, and updated associated figures and results. | 225 |
| NA | Many of the figure titles and labels in the supplemental need larger fonts. | 1 | Supplemental figures have been revised with larger font and sizing that improves visibility. | |
| 301 | It may be useful if the authors could describe why these measurement uncertainties were chosen. Also, is μmatm supposed to be μatm? | 1 | Our resasoning for these unceratinties was based on typical performance for field and lab measurements. We have added clarification to this section for these selections, with some references and fixed any typos. | 255 |
| 340-345 | How these uncertainty assessments were done are not clear. For example, the beginning of this paragraph refers to measurement uncertainty, but line 340 refers to uncertainty of monthly means. In the next sentence, the authors say that annual data from the WHOTS mooring are used to estimate uncertainty for HOT data, but it's unclear what "annual data" means when mooring data are 3-hourly and why moored pCO2 data would be used to determine uncertainty of HOT measurements of DIC. | 1 | We agree that this was not clearly communicated as written. The word annual was misleading here as it was referring to using a year of data. The point here though was to evaluate the uncertainty associated with averaging mooring data to monthly values as well as estimating the uncertainty associated with treating individual samples as monthly averages. WHOTS pCO2 data was used to estimate the daily variability in DIC at HOT and served as a proxy to estimate the uncertainty associated with monthly averaging. This combined with KEO and Papa provided a narrow range, from which we took the upper limit and applied to all sites. We have revised this section for clarity. | Section 2.7 |
| 655 | "lease" should likely be "'least" | 1 | done | |
| Fig 11 | One of the labels for the y axes on the right is cut off. | 1 | done | Fig 11 |
| Fig 14 | It's hard to tell the difference between the grays. Why not use the same color scheme as kernel density curves? | 1 | This figure was updated to match the color scheme used in other figures for visual consistency as suggested. | Fig 14 |
| NA | Overall I think the authors have done a very good job at responding to the issues raised by the reviewers. I'm happy to accept it in its present form, given the small issues highlighted below are amended. One preference I would still have though is for Figures 10 and 13 to present the anomalies, rather than leaving the reader to try and visualise them themselves. This is particularly as there are so many panels on the figures with so many lines on each panel - using the majority of the real estate for showing the observational variation doesn't appear to me to be the most efficient use of space. | 2 | Figures 10 and 13 were split into A and B parts and revised changed to residual plots as suggested. | |
| 16 | :...annual budgets [and] interannual and climatic variability" | 2 | done | 16 |
| 35 | "...over varied durations and may [be] trained with either in-situ..." | 2 | done | 35 |
| 152 and thro | but also throughout. When citing previous studies inline, I think the citation style should be "...from Lueker et al., (2000) rather than (Lueker, 2000)". Similarly "...Kf from (Dickson, 1979)..." should be replaced by "...Kf from Dickson (1979)..." | 2 | done | NA |
| 158 | should be (O'Reilly et al., 1998) | 2 | Fixed - this was and EndNote MS Word format typo (correctly referenced in library) | |
| 158-160 | - need citations for MODIS and VIIRS here | 2 | Citations and acknowledgements were added for MODIS and VIIRS data. | Section 2.2 |
| 189 | should be either 'measurement is' or 'measurements are' | 2 | done | |
| 257 | typo for absolute | 2 | done | |
| 296 | should be equation 10? | 2 | Equation numbering was fixed | |
| 337 | should be equation 11? | 2 | Equation numbering was fixed | |
| 339 | measurements rather than measurement | 2 | done | |
| 348 | should be equation 9 instead of equation 8? In fact, check all references to equations as these seem to be going awry up to this point. | 2 | Equation numbering was fixed | |
| 510 | Figure 3, could you explicitly put in the legend which colour is which? You can derive it from the differences in the trends of course, but it wouldn't hurt to include it. | 2 | Added legend here and additional visuallization to indicate time series were trunctated to Sept. 1997 coincident with remotely sensed chlorophyll records. | Fig. 3 |
| 567 | Kernel density curves. For readers that haven't come across these curves before, I'd recommend adding a line describing what they show, and what the optimal should be. (You have this in Fig 7 & 11 captions, just not in the text) | 2 | We added language here thatwas consistent with the captions for Figs. 7 & 11. | 361-362 |

| | | | | |
|---|---|---|---|---|
| 616 | Fig 10. There is a lot of information on this plot. Maybe have it over two pages, with three locations on each page? I still believe that anomalies from the observed of the timeseries would be more powerful and easier to interpret, showing the strengths of the different methods over different data gaps more clearly (this applies to the seasonal cycles shown in Fig 13 too, these are already shown in Fig 2 for observations, so showing anomalies from the observed for each imputation method would be easier to interpret). | 2 | Figures 10 and 13 were split into A and B parts and revised changed to residual plots as suggested. The composite time series versions from this revision were retained but moved into split 3-sited figures in the Supplemental Materials. | |
| 697 | typo of althought | 2 | done | |
| 881 | ... limited evaluation [of] errors… | 2 | done | |
| 899 | change 'less than' to 'less that' | 2 | done | |
| 965 | Should it be 'These were less than..' | 2 | done | |
| 968 | equating instead of equate | 2 | done | |
| 880 | It might be worth here stating what is thought to be the physical cause of the difference in performance given the different levels of missing data. Is it that the extremes of temperature and their DIC concentrations need to be captured so as to best enable the different imputation techniques (as temperature has the greatest correlation with DIC)? Or is it something else? | 2 | It is not clear if this comment / question is in reference to only the MLR performance across sites or the performance of each imputation model across sites, nor the distinction between physical causes for performance and missing levels of data. In either case, our interpretation of this comment would require additional analysis to appropriately answer without conjecture. | NA |
| 1133 | 'with acceptable accuracy' rather than 'with acceptable accurately' | 2 | done | |